# SENSE: SENSING SIMILARITY SEEING STRUCTURE

## ABSTRACT

Dimensionality reduction is widely used to visualize and analyze high-dimensional data, but most methods assume centralized access to all pairwise similarities, which is infeasible in privacy-sensitive, decentralized settings. We introduce **SENSE**, a geometry-aware framework for privacy-preserving decentralized representation learning. SENSE reconstructs global structure from sparse, locally observed distances via structured matrix completion, requiring no raw data sharing or iterative communication. It supports both Euclidean and hyperbolic geometries, adapts to flat and hierarchical structures, and operates under four deployment regimes reflecting real-world data availability. By design, SENSE safeguards raw features while producing faithful embeddings. Our theoretical analysis establishes formal privacy guarantees, and experiments on diverse benchmark datasets show that SENSE matches centralized baselines while remaining efficient and privacy-preserving. Our code is publicly available here.

## 1 INTRODUCTION

Dimensionality reduction (DR) projects high-dimensional data into lower-dimensional spaces where patterns are easier to interpret (Jolliffe & Cadima, 2016). A prominent family is neighbor embedding (NE) methods, which preserve local similarity relationships (Sorzano et al., 2014). Notable examples include t-SNE (van der Maaten & Hinton, 2008) and UMAP (McInnes et al., 2020), widely used in visualization (Cavallo & Demiralp, 2018), anomaly detection (Sadr et al., 2019), and exploratory analysis (Ding et al., 2002). Contrastive Neighbor Embedding (CNE) (Damrich et al., 2023) extends these ideas by casting neighborhood preservation into a contrastive learning framework that emphasizes the role of negatives. In contrast, PHATE (Moon et al., 2019) departs from NE altogether, leveraging diffusion geometry to capture both local and global structures. Together, these methods illustrate the diversity and impact of modern DR. However, most similarity-based DR approaches assume centralized access to all pairwise similarities, a condition rarely met in practice. In domains from healthcare and finance to IoT and social media, data are distributed across clients under strict privacy and communication constraints (Dwork et al., 2014; McMahan et al., 2017; Qiao et al., 2023), leaving inter-client similarities inaccessible. This fragmented view of the data yields embeddings that misrepresent relationships and fail to preserve the underlying structure (Li et al., 2024).

**Related Work.** Several approaches have addressed these challenges but are constrained by scalability, privacy, or deployment practicality. SMAP (Xia et al., 2020) offers strong privacy via Secure Multi-party Computation (SMC), but its cryptographic overhead makes large-scale use infeasible and prevents support for methods like UMAP. FedNE (Li et al., 2024) allows federated NE but lacks intrinsic privacy, relies on heavy server-client interaction, and is vulnerable to inversion attacks. Fed-tSNE and Fed-UMAP (Qiao et al., 2024) generate synthetic anchors via MMD alignment but assume multi-sample clients, fail in single-sample regimes, and remain susceptible to adversarial corruption. These limitations call for frameworks that are communication-efficient, privacy-preserving, and yield faithful embeddings. To this end, we propose SENSE, a geometry-aware framework for decentralized representation learning. It supports both Euclidean and hyperbolic spaces, the latter crucial for capturing hierarchical structures in domains such as social and biological networks (Malik et al., 2025). At its core, SENSE reconstructs global structure from sparse local distances, avoiding raw data sharing, iterative communication, or centralized storage. The completed distance matrix is then used with classical NE methods, CNE, PHATE, or hyperbolic CoSNE (Guo et al., 2022), enabling scalable and privacy-preserving embeddings in decentralized environments.

The discussion so far has emphasized privacy as a key barrier to decentralized representation learning, but this naturally raises a deeper question: *what do we actually mean by privacy*? The term is

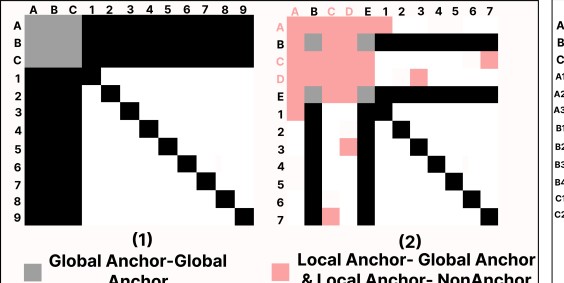 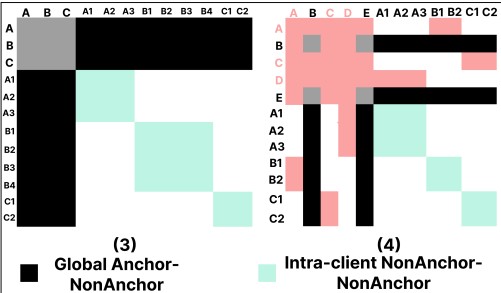

Figure 1: Observed entries in the global distance matrix $D$ under four SENSE configurations: (1) *Pointwise-Full*, (2) *Pointwise-Partial*, (3) *Multisite-Full*, and (4) *Multisite-Partial*. Configurations differ by visibility of Anchor–NonAnchor (A–NA) and NA–NA blocks, determined by client data locality and anchor access. In *Pointwise*, each client contributes a single NA (e.g., 1, 2, ..., 9), whereas *Multisite* allows intra-client NA–NA observations (e.g., A1, A2, ..., C2). *Full* modes grant all NAs access to the global anchor set (e.g., A–E), yielding complete A–NA blocks, while *Partial* modes restrict clients to disjoint anchor subsets, producing sparse structured observations.

inherently contextual. Across domains such as healthcare, finance, and social platforms, data often encode highly sensitive attributes such as medical records, geolocation traces, genomic records, financial transactions, and social interactions (Dwork et al., 2014; Rieke et al., 2020; Byrd & Polychroniadou, 2020; Lim et al., 2020). Over the years, privacy has been formalized in diverse ways, from cryptographic guarantees to statistical indistinguishability and database anonymity, reflecting that there is no single universal notion, only definitions shaped by the threat model and application context (Yao, 1982; Goldreich, 1998; Dwork et al., 2006; Abadi et al., 2016; Sweeney, 2002; Kairouz et al., 2021). So, *what does privacy mean in our setting*?

*In* SENSE*, clients may typically hold sensitive datasets, such as patient health records, demographic attributes, or financial profiles, making the raw feature vectors $\{x_i\}$ the sensitive objects. Disclosing such data would be a severe violation. Instead,* SENSE *relies on distances to some anchor points as safe coordination signals, reconstructing inter-client similarity (i.e., who is close to whom) without exposing raw features, thereby producing faithful low-dimensional embeddings under strict privacy and communication constraints.*

Anchors therefore play a central role in our framework, and their use is both practical and robust. Curated by a trusted server, they can be synthetic, anonymized, or drawn from public data, decoupled from private client records. This avoids leakage risks from client-generated anchors, especially in small or skewed regimes (Qiao et al., 2024) while providing stability, auditability, and adversarial robustness. Such strategies are already common in healthcare (Johnson et al., 2016; Bycroft et al., 2018), genomics (Regev et al., 2017; Litviňuková et al., 2020), finance (Awosika et al., 2024a), mobile/NLP applications (Hard et al., 2019; Li et al., 2019), and wireless sensor networks (Di Franco et al., 2017), illustrating their practical viability. Motivated by this, we treat anchors as core architectural components. SENSE leverages them, together with tools from distance matrix completion, network localization, and low-rank recovery, to provide formal privacy guarantees for reconstructing global structure from partial observations. It introduces the following key innovations:

- *Privacy and communication efficiency:* Distance estimation is decoupled from raw data, avoiding reliance on external mechanisms such as Homomorphic Encryption or Differential Privacy. Also, it requires only a single client–server interaction with no iterative training.
- *Geometric and deployment flexibility:* Supports both Euclidean and hyperbolic spaces and adapts to four observation regimes. As shown in Figure 1, these regimes dictate which entries of the distance matrix are observable. SENSE estimates/infers the missing inter-client similarity from this incomplete information while preserving privacy across all scenarios. In the *Pointwise* setting, each client contributes only a single non-anchor (NA), typical of edge/mobile devices, whereas in the *Multisite* setting, clients hold multiple NAs, such as patients in hospitals or customers in banks.
- *Provable reliability:* Provides formal privacy guarantees, complemented by empirical validation across diverse datasets and geometries.

**Practical Impact.** These properties make SENSE broadly applicable to privacy-sensitive, structurally diverse domains. Hospitals can jointly visualize patient data without violating HIPAA/GDPR (Sheller et al., 2019), banks can detect fraud patterns without exposing transac-

tions (Awosika et al., 2024b), and even mobile/IoT devices with a single sample can contribute to global embeddings (Pape & Rannenberg, 2019; Baran, 1964). Genomic labs can embed single-cell transcriptomes into a shared hyperbolic space that preserves both cellular hierarchy and privacy (Agnihotry et al., 2022; Tasissa & Lai, 2019). Crucially, SENSE also supports dynamic participation: new clients or samples can be incorporated by estimating partial distances to existing entities, avoiding full re-computation while maintaining global coherence. Thus, SENSE is not only privacy-preserving and geometry-aware but also inherently scalable to dynamic federated ecosystems.

## 2 BACKGROUND AND PROBLEM FORMULATION.

**Neighbor Embedding (NE).** Methods like t-SNE (van der Maaten & Hinton, 2008) and UMAP (Damrich & Hamprecht, 2021) embed high-dimensional data $\mathbf{X} = \{x_i\}_{i=1}^n \subset \mathbb{R}^{d_h}$ into a low-dimensional space $\mathbf{Y} = \{y_i\}_{i=1}^n \subset \mathbb{R}^{d_\ell}$ by preserving pairwise structure. These methods are distance-driven. They transform distances into similarities via kernels to preserve relational structure (see Appendix A.1, A.2). Let $\mathbf{D}_{ij}^{d_h} = \|x_i - x_j\|$ and $\mathbf{D}_{ij}^{d_\ell} = \|y_i - y_j\|$ denote distances in the high- and low-dimensional spaces respectively. These are mapped to similarities via kernel functions: $S_{ij}^{d_h} = f(\mathbf{D}_{ij}^{d_h}), \quad S_{ij}^{d_\ell} = g(\mathbf{D}_{ij}^{d_\ell})$, where $f$ and $g$ are typically Gaussian, Laplacian, or Cauchy kernels. The general NE objective minimizes the divergence between the two similarity matrices:

$$\mathcal{L}(\mathbf{Y}) = \sum_{i,j} \mathcal{D}(S_{ij}^{d_h}, S_{ij}^{d_\ell}), \tag{1}$$

where $\mathcal{D}$ is a divergence measure such as KL divergence or binary cross-entropy.

**Contrastive Neighbor Embedding.** CNE (Damrich et al., 2023) extends NE into the contrastive learning framework by training an encoder $f_\theta$ to map $\mathbf{x}_i$ to $\mathbf{y}_i = f_\theta(\mathbf{x}_i)$ such that the neighborhood structure from a $k$-NN graph is preserved (Li et al., 2024). CNE uses a distance-aware contrastive loss (see Def A.3 in Appendix), framed as a binary similarity matching problem. Let $S^{d_h} \in \{0,1\}^{n \times n}$ denote ground-truth neighborhood indicators and $S^{d_l}$ denote kernel-based similarities in the embedding space. The loss is a weighted binary cross-entropy:

$$\mathcal{L}(\mathbf{Y}) = -\sum_{i,j} \left[ S_{ij}^{d_h} \log S_{ij}^{d_l} + b(1 - S_{ij}^{d_h}) \log(1 - S_{ij}^{d_l}) \right]. \tag{2}$$

where $b > 0$ balances the repulsion term; for more details on Eq. 2 see A.3.

*Key Challenges in Decentralized Settings.* **(C1)** CNE, like NE, relies on a full similarity matrix, which is unavailable in privacy-sensitive, decentralized settings. **(C2)** Conventional distributed learning captures only intra-client structure, omitting crucial inter-client neighbor information. **(C3)** Clients lack access to global data, leading to incorrect kNN graphs and biased negative sampling, as true neighbors may reside on other clients.

**CO-SNE (for Hyperbolic Data).** Hierarchical structures in social, biological, and knowledge graphs grow exponentially, making Euclidean embeddings unsuitable due to distortion of tree-like geometry. Hyperbolic space, with constant negative curvature, naturally models such growth and supports hierarchy-aware learning (Malik et al., 2025; Ganea et al., 2018) (see Appendix A.3.1). Standard methods like t-SNE assume Euclidean geometry and distort global structure when applied to hyperbolic data, collapsing depth and relative positioning. CO-SNE (Guo et al., 2022) extends t-SNE to hyperbolic geometry by using distance-aware kernels: $S_{ij}^{d_h} = f(d_{\mathbb{B}^n}(x_i, x_j)), \quad S_{ij}^{d_l} = g(d_{\mathbb{B}^2}(y_i, y_j))$, where $f$ is a hyperbolic normal kernel and $g$ a heavy-tailed Cauchy kernel. A depth-regularization term aligns norms across spaces. The objective is:

$$\mathcal{L}(\mathbf{Y}) = \lambda_1 \cdot \mathcal{D}(S^{d_h}, S^{d_l}) + \lambda_2 \sum_i (\rho(x_i) - \rho(y_i))^2, \tag{3}$$

where $\rho(x) = \|x\|$ and $\mathcal{D}$ is typically KL divergence. For more details see Def A.4.

### 2.1 PROBLEM FORMULATION

We consider a decentralized system with $M$ clients $\{\mathcal{C}_1, \ldots, \mathcal{C}_M\}$ coordinated by a central server owned by a private company, hospital, bank, or government agency. Each client $\mathcal{C}_m$ holds a private dataset $\mathcal{D}_m = \{\mathbf{x}_i^m\}_{i=1}^{N_m} \subset \mathbb{R}^{d_h}$, which remains local and disjoint, i.e., $\mathcal{D}_m \cap \mathcal{D}_{m'} = \emptyset$ for $m \neq m'$. Let $N = \sum_{m=1}^M N_m$ be the total number of data points, indexed globally by $i \in [N]$. We consider two real-world configurations: A) *SENSE-Pointwise*, where each client holds a single sample $\mathbf{x}^m \in \mathbb{R}^{d_h}$,

and B) *SENSE-Multisite*, where each client holds a local dataset $\mathbf{X}^m = [\mathbf{x}_1^m, \ldots, \mathbf{x}_{N_m}^m] \in \mathbb{R}^{N_m \times d_h}$. Let $\mathbf{D} \in \mathbb{R}^{N \times N}$ denote the full squared distance matrix. In Euclidean space, $\mathbf{D}_{ij} = \|\mathbf{x}_i - \mathbf{x}_j\|^2$; in hyperbolic space, it reflects squared distances in the Poincaré ball $\mathbb{B}^{d_h}$ or Lorentz model $\mathbb{H}^{d_h}$ (see Appendix A.3). Due to privacy constraints, only a subset of entries is observable. Let $\Omega \subseteq [N] \times [N]$ be the set of observed indices, and define the projection operator $\mathcal{P}_\Omega : \mathbb{R}^{N \times N} \to \mathbb{R}^{N \times N}$ as:

$$[\mathcal{P}_\Omega(\mathbf{D})]_{ij} = \begin{cases} \mathbf{D}_{ij}, & \text{if } (i,j) \in \Omega, \\ 0, & \text{otherwise.} \end{cases} \tag{4}$$

**Goal 1** *Our goal is to reconstruct the full distance matrix from partial observations $\mathbf{D}_\Omega = \mathcal{P}_\Omega(\mathbf{D})$ via structured matrix completion. Rather than estimating distances directly, we infer latent embeddings $\widehat{\mathbf{X}} \in \mathbb{R}^{N \times d_h}$ whose induced distances agree with the observed entries. Formally, we solve:*

$$\widehat{\mathbf{X}}^\star = \arg\min_{\widehat{\mathbf{X}}} \left\| \mathcal{P}_\Omega\left(\mathcal{D}(\widehat{\mathbf{X}})\right) - \mathbf{D}_\Omega \right\|_F^2, \tag{5}$$

*and define the reconstructed distance matrix as $\widehat{\mathbf{D}} = \mathcal{D}(\widehat{\mathbf{X}}^\star)$, which serves as an approximation of the true but unknown $\mathbf{D}$. Here $\mathcal{D}(\widehat{\mathbf{X}})$ denotes the induced pairwise distance matrix under the chosen geometry (Euclidean or hyperbolic). From $\widehat{\mathbf{D}}$, we then derive a global low-dimensional embedding $\mathbf{Y} = \{\mathbf{y}_i\}_{i=1}^N \subset \mathbb{R}^{d_\ell}$ with $d_\ell \ll d_h$, which preserves the neighborhood structure.*

We use $\widehat{\mathbf{D}}$ to find the similarities, defined in Eq. 6 and optimized via divergence $\mathcal{D}(S^{d_h}, S^{d_\ell})$ (Eq. 1).

$$S_{ij}^{d_h} = \exp\left(-\frac{\widehat{\mathbf{D}}_{ij}}{2\sigma^2}\right), \quad S_{ij}^{d_\ell} = g(\|\mathbf{y}_i - \mathbf{y}_j\|^2), \tag{6}$$

For contrastive learning, we build binary similarities using $k$-nearest neighbors:

$$S_{ij}^{d_h} = \begin{cases} 1, & \text{if } j \in \text{kNN}(i; \widehat{\mathbf{D}}), \\ 0, & \text{otherwise,} \end{cases} \quad S_{ij}^{d_\ell} = \phi(\mathbf{y}_i, \mathbf{y}_j) = \frac{1}{1 + \|\mathbf{y}_i - \mathbf{y}_j\|^2}, \tag{7}$$

and minimize the contrastive loss (Eq. 2). For hierarchical data, we apply CO-SNE, treating $\widehat{\mathbf{D}}$ as squared hyperbolic distances in the Poincaré model to compute similarities (Eq. 16 in Appendix). The embedding $\mathbf{Y} \subset \mathbb{B}^{d_\ell}$ is optimized using the CO-SNE loss (Eq. 3).

## 3 PROPOSED FRAMEWORK: SENSE

As described in Section 2.1, we consider two decentralized settings: *SENSE-Pointwise* and *SENSE-Multisite*. In both, each client holds private non-anchor (NA) data and accesses a shared anchor set $\mathcal{A} = \{a_1, \ldots, a_K\}$ with feature matrix $\mathbf{X}_A = [\mathbf{p}_1, \ldots, \mathbf{p}_K]^\top \in \mathbb{R}^{K \times d_h}$. Anchors, broadcast by the server, may be global or client-specific (see Appendix A.8). Let $\mathcal{X} = \{x_1, \ldots, x_N\}$ be the set of all private NA points (raw features), where $N = \sum_{m=1}^M N_m$. Each client computes squared distances between its NAs and accessible anchors:

$$\mathbf{d}_i^m = \left[ \|x_i^m - \mathbf{p}_1\|^2, \ldots, \|x_i^m - \mathbf{p}_K\|^2 \right],$$

and transmits these to the server, masking unshared local anchors. In *Pointwise*, each client contributes one NA-anchor vector, in *Multisite*, intra-client NA–NA distances may also be known. The global incomplete squared distance matrix $\mathbf{D} \in \mathbb{R}^{(K+N) \times (K+N)}$ is partitioned as:

$$\mathbf{D} = \begin{bmatrix} E & F \\ F^\top & G \end{bmatrix}, \tag{8}$$

where $E$ is anchor–anchor, $F$ is anchor–NA, and $G$ is NA–NA. The observed subset is indexed by $\Omega \subseteq [K+N]^2$, based on anchor visibility and client configuration. We consider four configurations: *Pointwise-Full*, *Pointwise-Partial*, *Multisite-Full*, and *Multisite-Partial* which differ in the extent of observed entries in $F$ (anchor–NA) and $G$ (NA–NA). These define distinct visibility patterns in $\Omega$, summarized in Appendix Table 6 and illustrated in Figure 1, and determine which distances are available for structured matrix completion. To reconstruct the unobserved blocks of $\mathbf{D}$ (notably $G$) and obtain the *reconstructed* matrix $\widehat{\mathbf{D}}$ (or $\widehat{G}$) we use geometry-specific solvers: anchored-MDS in Euclidean space (Sec. 3.1) and LHYDRA (Keller-Ressel & Nargang, 2022) in hyperbolic space. The complete pipeline is outlined in Algorithm 1 (Appendix).

**Remark 1** *In practice, $F$ may be only partially or fully visible due to bandwidth, privacy, or data limitations. SENSE is designed to operate under such conditions.*

### 3.1 SENSE via Anchored-MDS

Classical MDS embeds $N$ points by minimizing stress over a fully observed distance matrix $\mathbf{D} \in \mathbb{R}^{N \times N}$. The embedding $\mathbf{X} \in \mathbb{R}^{N \times d_h}$ minimizes: $\sigma(\mathbf{X}) = \sum_{i<j} (\|x_i - x_j\| - \delta_{ij})^2$, where $\delta_{ij}$ is the input Euclidean distance between points $i$ and $j$. SMACOF solves this using a majorization-based surrogate (De Leeuw, 2005) (details in Appendix A.5), $\tau(\mathbf{X}, \mathbf{Z}) = C + \mathrm{tr}(\mathbf{X}^\top \mathbf{V} \mathbf{X}) - 2\,\mathrm{tr}(\mathbf{X}^\top \mathbf{B}(\mathbf{Z})\mathbf{Z})$, with the iterative update:

$$\mathbf{X}^{(k)} = \mathbf{V}^\dagger \mathbf{B}(\mathbf{X}^{(k-1)})\mathbf{X}^{(k-1)}. \tag{9}$$

In SENSE, we do not observe the full matrix $\mathbf{D}$ (in Eq. 8), instead, we only access the observed entries $\mathbf{D}_\Omega = \mathcal{P}_\Omega(\mathbf{D})$, which contain distances on a subset of pairs. Let the embedding be $\mathbf{X} = [\mathbf{X}_A\ \mathbf{X}_{NA}]^\top$, where $\mathbf{X}_A$ and $\mathbf{X}_{NA}$ are anchor and NA embeddings, respectively. Stress is minimized over observed entries only: $\sigma(\mathbf{X}) = \|\mathcal{P}_\Omega(\mathcal{D}(\mathbf{X}) - \mathbf{D})\|_F^2$, where $\mathcal{P}_\Omega$ projects onto observed indices $\Omega$, and $\mathcal{D}(\mathbf{X})$ computes pairwise distances. The SMACOF updates are restricted to $\Omega$, with:

$$V_{ij} = \begin{cases} |\{j : (i,j) \in \Omega\}|, & i = j \\ -1, & (i,j) \in \Omega,\ i \neq j \\ 0, & \text{otherwise} \end{cases} \quad B_{ij}(\mathbf{X}) = \begin{cases} -\frac{\delta_{ij}}{\|x_i - x_j\|}, & (i,j) \in \Omega,\ i \neq j \\ -\sum_{k \neq i,\ (i,k) \in \Omega} B_{ik}, & i = j \\ 0, & \text{otherwise} \end{cases}$$

We partition $V$ and $B$ as defined in Eq. 10, where $V_{AA}, B_{AA} \in \mathbb{R}^{K \times K}$, $V_{AN}, B_{AN} \in \mathbb{R}^{K \times N}$, and $V_{NN}, B_{NN} \in \mathbb{R}^{N \times N}$:

$$\mathbf{V} = \begin{bmatrix} \mathbf{V}_{AA} & \mathbf{V}_{AN} \\ \mathbf{V}_{AN}^\top & \mathbf{V}_{NN} \end{bmatrix}, \quad \mathbf{B} = \begin{bmatrix} \mathbf{B}_{AA} & \mathbf{B}_{AN} \\ \mathbf{B}_{AN}^\top & \mathbf{B}_{NN} \end{bmatrix} \tag{10}$$

The update rule for NA embeddings becomes:

$$\mathbf{X}_{NA}^{(k)} = \mathbf{V}_{NN}^\dagger \left( \mathbf{B}_{NN} \mathbf{X}_{NA}^{(k-1)} + \mathbf{B}_{AN}^\top \mathcal{P}_\Omega(\mathbf{X}_A) - \mathbf{V}_{AN}^\top \mathcal{P}_\Omega(\mathbf{X}_A) \right). \tag{11}$$

This projection-aware update ensures $\mathbf{X}_{NA}$ uses only observed/available distances. The projection operator $\mathcal{P}_\Omega$ acts as a binary mask over observed entries. While $\mathbf{V}$ and $\mathbf{B}$ are derived from $\Omega$, we apply $\mathcal{P}_\Omega$ to $\mathbf{X}_A$ in Eq. 11 to retain only anchors with observed anchor–NA distances. This avoids leakage from inaccessible anchors and ensures privacy-compliant updates (pseudocode in Appendix A.7). Furthermore, to preserve privacy, the number of shared anchors $K$ must be limited. Theorems 3.1, 3.2 (Euclidean) and Lemma 1 (hyperbolic) characterize how $K$ relates to embedding dimension $d_h$ across SENSE configurations, establishing privacy conditions for faithful reconstruction.

**Theorem 3.1** *Let $\mathcal{X} = \{\mathbf{x}_1, \ldots, \mathbf{x}_N\} \subset \mathbb{R}^{d_h}$ be the set of NA data points, and let $\mathcal{A} = \{\mathbf{a}_1, \ldots, \mathbf{a}_K\} \subset \mathbb{R}^{d_h}$ be the set of $K$ anchor points. Suppose we observe the pairwise Euclidean distances $\{\|\mathbf{x}_i - \mathbf{a}_j\|\}_{i \in [N], j \in [K]}$ between each NA and all anchors. If the number of anchors satisfies $K < d_h$, then the original NA features $\{\mathbf{x}_i\}_{i=1}^N$ cannot be exactly reconstructed from these distances, guaranteeing the privacy of the individual client data.*

*Proof.* Deferred in Appendix, check A.1.

While Theorem 3.1 ensures privacy, the reconstructed embeddings (Eq. 11) must also remain useful. We capture this through *reconstruction fidelity*, defined as preserving neighborhood structure rather than exact features. In SENSE, such fidelity is guaranteed by well-established results from MDS distance-based recovery (Drineas et al., 2006; Zhang et al., 2019; Lichtenberg & Tasissa, 2024a). These solvers are known to produce embeddings consistent up to Euclidean isometries (translation, rotation, and scaling) (Mardia & Riley, 2021; Khan et al., 2009). These invariances ensure that geometric relationships are preserved for downstream tasks, while exact feature values remain unrecoverable, achieving precisely the balance required in privacy-sensitive settings.

SENSE supports multiple configurations, which critically influence embedding fidelity and privacy. Theorem 3.2 formalizes privacy guarantees when only partial anchor–NA distances (block $F$) are available, covering both *pointwise* and *multisite* regimes. *1) SENSE-Pointwise:* Each client $j \in [N]$ holds a single private point $\boldsymbol{x}_j \in \mathbb{R}^{d_h}$ and accesses a subset of anchors indexed by $\mathcal{I}_j \subseteq [K]$. The corresponding anchor set is $\mathcal{A}_j = \{\boldsymbol{a}_i\}_{i \in \mathcal{I}_j}$, comprising: (i) global anchors $\mathcal{A}_G = \{\boldsymbol{a}_1, \ldots, \boldsymbol{a}_{M_G}\}$, shared across all clients, and (ii) local anchors $\mathcal{A}_L^{(j)}$, unique to client $j$. The total of anchors observed is $r_j = |\mathcal{I}_j| = M_G + M_L^{(j)}$. *2) SENSE-Multisite:* Each client $m \in [M]$ holds a local dataset $\mathcal{X}^{(m)} = \{\boldsymbol{x}_{m,1}, \ldots, \boldsymbol{x}_{m,n_m}\} \subset \mathbb{R}^{d_h}$, where $N = \sum_{m=1}^M n_m$. Each point $\boldsymbol{x}_{m,i}$ observes distances

to (i) a shared global anchor set $\mathcal{A}_G$, and (ii) a local anchor set $\mathcal{A}_L^{(m)}$ exclusive to client $m$. Let $\mathcal{I}_{m,i} = \mathcal{I}_G \cup \mathcal{I}_L^{(m)}$ be index set of accessible anchors, with $r_{m,i} = |\mathcal{I}_{m,i}|$ denoting number observed.

**Theorem 3.2** *Let $\mathcal{X} = \{\boldsymbol{x}_1, \ldots, \boldsymbol{x}_N\} \subset \mathbb{R}^{d_h}$ be the set of all non-anchor (NA) points across all clients, where each $\boldsymbol{x}_i$ computes squared distances only to a subset of accessible anchors $\mathcal{A}_i = \{\boldsymbol{a}_j\}_{j \in \mathcal{I}_i}$, with $|\mathcal{I}_i| = r_i$. If $r_i < d_h$ for all $i \in [N]$, then exact recovery of each $\boldsymbol{x}_i$ is impossible. The inverse map from anchor distances to features is non-unique, preserving privacy under both pointwise and multisite configurations.*

*Proof.* Deferred in Appendix, check A.2.

**Lemma 1** *Let $\{x_1, \ldots, x_{K+N}\} \subset \mathbb{H}^{d_h}$ be $K$ anchors and $N$ NA points in hyperbolic space with curvature $-\kappa$. Suppose only blocks $E$ and $F$ of distance matrix $\mathbf{D}$ are observed. If $K < d_h$, the NA coordinates cannot be exactly recovered up to isometry in $\mathbb{H}^{d_h}$, ensuring the privacy of the client data in SENSE. This follows from the contrapositive of the L-HYDRA theorem (Keller-Ressel & Nargang, 2022), which guarantees exact recovery only when $K \geq d_h$ and anchors span a full subspace.*

In SENSE, we deliberately restrict anchors to $K < d_h$ rather than $K \leq d_h$. Fewer anchors enlarge the feasible solution space, introducing geometric ambiguity (Wei et al., 2015; Liberti et al., 2014) that strengthens privacy while still preserving neighborhood structure (Fig. 3). This design holds in both Euclidean and hyperbolic spaces, making it a general, geometry-agnostic choice. A detailed theoretical discussion and supporting examples are provided in Appendix A.10.

## 4 EXPERIMENTS

In this section, we first outline the experimental setup, followed by an evaluation of SENSE across diverse datasets and deployment settings.

### 4.1 EXPERIMENTAL SETUP

**Datasets.**    We evaluate SENSE on 14 public datasets widely used in DR and representation learning (Fu et al., 2024). These include three benchmarks: MNIST (Deng, 2012), Fashion-MNIST (Xiao et al., 2017), and CIFAR-10 (Giuste & Vizcarra, 2020); seven MedMNIST datasets (Yang et al., 2023): DermaMNIST, PneumoniaMNIST, RetinaMNIST, BreastMNIST, BloodMNIST, OrganCMNIST, OrganSMNIST; and the German Credit dataset (Hofmann, 1994). For hyperbolic, we use three graph datasets: Airport (Malik et al., 2025), Amazon (Yang & Leskovec, 2012), and DBLP (Kataria et al., 2024). Detailed dataset stats and system specifications are in Appendix Table 7 and A.13.

**Baselines.**    We compare SENSE against centralized (Van) baselines: t-SNE (van der Maaten & Hinton, 2008), UMAP (McInnes et al., 2020), PHATE (Moon et al., 2019), and CNE (Damrich et al., 2023) (with $s \in \{0, 0.5, 1\}$). These assume full raw data access at a central server and serve as upper bounds for evaluating SENSE's privacy-preserving performance.

**Implementation Details.**    SENSE has two stages: matrix completion and global embedding. In the first stage, data is partitioned across $M$ clients. In *Pointwise*, each client holds one NA point, sampled randomly. In *Multisite*, clients hold multiple NA points under IID or non-IID splits (balanced/unbalanced). A subset of $10\%$ of the total data points is designated as anchors. In *Full* settings, all anchors are global, and in *Partial*, anchors are split into global and client-specific local sets. The total of anchors (global + local) is fixed at $d_h - 1$, where $d_h$ is the original feature dimension. In the embedding stage, we use the completed distance matrix to generate privacy-preserving embeddings using multiple neighbor embedding methods. For Euclidean geometry, we use the official implementations of t-SNE (van der Maaten & Hinton, 2008), UMAP (McInnes et al., 2020), and PHATE (via its standard Python library). For CNE, we adopt the implementation from (Damrich et al., 2023), where the parameter $s$ controls the attraction-repulsion tradeoff: $s = 0$ mimics t-SNE, $s = 1$ aligns with UMAP, and intermediate values interpolate between them. CNE operates within a contrastive learning framework using negative sampling. For hyperbolic embeddings, we use the CO-SNE implementation from (Guo et al., 2022).

**Remark 2** *The $10\%$ anchor sharing in the multisite setting is used only for empirical evaluation. These anchors are not private; they act as public or semi-public landmarks, akin to those in GPS (Shang & Ruml, 2004) or radar systems (Iannucci et al., 2020). This is standard in localization literature (Di Franco et al., 2017; Khan et al., 2009), where landmarks aid positioning but are not privacy-sensitive (Koledoye et al., 2017). Our privacy definition protects only the high-dimensional features of NA points. Anchors are fixed, visible, and either synthetic, public, or explicitly consented. Not part of any client's private data. For details on anchor generation, see Appendix A.8.*

Table 1: Full vs. Partial comparison in MULTISITE under non-IID (unbalanced) splits. Evaluation spans centralized and privacy-preserving SENSE variants across different embedding quality metrics.

| Data | Metric | t-SNE | | UMAP | | PHATE | | CNE(s=0) | | CNE(s=0.5) | | CNE(s=1) | |
|---|---|---|---|---|---|---|---|---|---|---|---|---|---|
| | | VAN. | SENSE | VAN. | SENSE | VAN. | SENSE | VAN. | SENSE | VAN. | SENSE | VAN. | SENSE |
| | | | | | | — Multisite-Partial Setting — | | | | | | | |
| MNIST | Trust. | 0.9890 | 0.9898 | 0.9553 | 0.9552 | 0.8741 | 0.8763 | 0.9517 | 0.9521 | 0.9524 | 0.9538 | 0.9455 | 0.9476 |
| | Cont. | 0.9575 | 0.9639 | 0.9774 | 0.9771 | 0.9811 | 0.9804 | 0.9806 | 0.9797 | 0.9799 | 0.9787 | 0.9799 | 0.9787 |
| | Stead. | 0.7719 | 0.7861 | 0.7639 | 0.7635 | 0.6628 | 0.6746 | 0.7840 | 0.7790 | 0.7752 | 0.7768 | 0.7634 | 0.7658 |
| | Cohes. | 0.8189 | 0.8458 | 0.8865 | 0.8853 | 0.8668 | 0.8877 | 0.9229 | 0.9112 | 0.9107 | 0.9196 | 0.9158 | 0.9087 |
| fashionMNIST | Trust. | 0.9902 | 0.9914 | 0.9140 | 0.9148 | 0.9579 | 0.9557 | 0.9765 | 0.9752 | 0.9784 | 0.9769 | 0.9765 | 0.9731 |
| | Cont. | 0.9608 | 0.9590 | 0.9812 | 0.9818 | 0.9910 | 0.9906 | 0.9915 | 0.9913 | 0.9905 | 0.9903 | 0.9900 | 0.9901 |
| | Stead. | 0.8415 | 0.8643 | 0.7570 | 0.7622 | 0.7836 | 0.7891 | 0.8632 | 0.8638 | 0.8643 | 0.8660 | 0.8493 | 0.8513 |
| | Cohes. | 0.6496 | 0.6559 | 0.6748 | 0.7069 | 0.7051 | 0.7115 | 0.7680 | 0.7669 | 0.7637 | 0.7508 | 0.7792 | 0.7666 |
| | | | | | | — Multisite-Full Setting — | | | | | | | |
| MNIST | Trust. | 0.9890 | 0.9852 | 0.9553 | 0.9570 | 0.8741 | 0.8780 | 0.9517 | 0.9516 | 0.9524 | 0.9542 | 0.9455 | 0.9452 |
| | Cont. | 0.9575 | 0.9518 | 0.9774 | 0.9754 | 0.9811 | 0.9797 | 0.9806 | 0.9772 | 0.9799 | 0.9763 | 0.9799 | 0.9761 |
| | Stead. | 0.7719 | 0.7953 | 0.7639 | 0.7726 | 0.6628 | 0.6688 | 0.7840 | 0.7808 | 0.7752 | 0.7828 | 0.7634 | 0.7690 |
| | Cohes. | 0.8189 | 0.8328 | 0.8865 | 0.8665 | 0.8668 | 0.8818 | 0.9229 | 0.9047 | 0.9107 | 0.8926 | 0.9158 | 0.9106 |
| fashionMNIST | Trust. | 0.9902 | 0.9895 | 0.9140 | 0.9076 | 0.9579 | 0.9555 | 0.9765 | 0.9752 | 0.9784 | 0.9769 | 0.9765 | 0.9725 |
| | Cont. | 0.9608 | 0.9731 | 0.9812 | 0.9797 | 0.9910 | 0.9902 | 0.9915 | 0.9906 | 0.9905 | 0.9895 | 0.9900 | 0.9891 |
| | Stead. | 0.8415 | 0.8604 | 0.7570 | 0.7530 | 0.7836 | 0.7981 | 0.8632 | 0.8608 | 0.8643 | 0.8649 | 0.8493 | 0.8538 |
| | Cohes. | 0.6496 | 0.6936 | 0.6748 | 0.7019 | 0.7051 | 0.7039 | 0.7680 | 0.7503 | 0.7637 | 0.7591 | 0.7792 | 0.7695 |
| | | | | | | — Pointwise-Full Setting — | | | | | | | |
| MNIST | Trust. | 0.9661 | 0.9679 | 0.9484 | 0.9467 | 0.8457 | 0.8469 | 0.9218 | 0.9166 | 0.9164 | 0.9138 | 0.9137 | 0.9151 |
| | Cont. | 0.9418 | 0.9410 | 0.9376 | 0.9396 | 0.9546 | 0.9538 | 0.9434 | 0.9422 | 0.9428 | 0.9417 | 0.9409 | 0.9403 |
| | Stead. | 0.8083 | 0.8113 | 0.7878 | 0.7763 | 0.6953 | 0.6958 | 0.8024 | 0.8003 | 0.8041 | 0.7996 | 0.8025 | 0.7914 |
| | Cohes. | 0.7904 | 0.7998 | 0.7855 | 0.7819 | 0.7912 | 0.7843 | 0.7988 | 0.7982 | 0.8034 | 0.7894 | 0.7931 | 0.7919 |
| fashionMNIST | Trust. | 0.9647 | 0.9681 | 0.9441 | 0.9434 | 0.8407 | 0.8375 | 0.9283 | 0.9264 | 0.9255 | 0.9245 | 0.9256 | 0.9196 |
| | Cont. | 0.9430 | 0.9454 | 0.9386 | 0.9373 | 0.9542 | 0.9528 | 0.9464 | 0.9460 | 0.9456 | 0.9440 | 0.9451 | 0.9429 |
| | Stead. | 0.8118 | 0.8103 | 0.7797 | 0.7779 | 0.6923 | 0.6931 | 0.8087 | 0.8049 | 0.8085 | 0.8003 | 0.8082 | 0.8150 |
| | Cohes. | 0.7570 | 0.7882 | 0.7685 | 0.7670 | 0.7564 | 0.7599 | 0.7876 | 0.7786 | 0.7843 | 0.7788 | 0.7838 | 0.7710 |

**Data Partitioning.** To simulate realistic distributed settings, we evaluate SENSE under both IID and non-IID distributions using Dirichlet-based partitioning. For each class $c$, client-wise proportions are drawn from $q_c \sim \text{Dir}(\alpha)$, where lower $\alpha$ yields greater heterogeneity and class imbalance (Wang et al., 2020; Zhao et al., 2018). We set $\alpha = 0.5$ in all experiments. Three partitioning schemes are used: *IID* (uniform class mix), *non-IID balanced* (varying class distributions, equal client sizes), and *non-IID unbalanced* (both class and size vary).

**Evaluation Metrics.** We assess SENSE using both reconstruction and embedding quality metrics. For fidelity, we compute *Relative Distance Error (DE)* and *F-score (FS)* between the reconstructed distance matrix (NA-NA) $\hat{G}$ and ground truth $G_{\text{true}}$: $\text{DE} = \frac{\|\hat{G} - G_{\text{true}}\|_F}{\|G_{\text{true}}\|_F}$, and $\text{FS} = \frac{2\,\text{tp}}{2\,\text{tp} + \text{fp} + \text{fn}}$, where tp, fp, and fn are true, false positive, and false negative neighbors respectively (Egilmez et al., 2017). To evaluate 2D embeddings, we compute *Trustworthiness* and *Continuity* (Venna & Kaski, 2005), which measure neighborhood agreement between original and embedded spaces. We also report *Steadiness* and *Cohesiveness* (Jeon et al., 2021) to assess global structural reliability: steadiness detects false groupings and cohesiveness quantifies how well true input clusters are preserved.

### 4.2 RESULT ANALYSIS.

We comprehensively evaluate SENSE across: *1) Standard image datasets (MNIST, FashionMNIST, CIFAR-10):* These are evaluated under *Pointwise-Full*, *Multisite-Full*, and *Multisite-Partial* with non-IID unbalanced splits. As shown in Table 1 and in Appendix 10, SENSE closely matches centralized baselines across Cont., Trust., Stead., and Cohes. Notably, the *Partial* configuration performs comparably to *Full*, indicating that accurate reconstruction of the global distance matrix is possible even with partial anchor–NA observations. Table 9 further confirms high F-score and low distance error, validating strong neighborhood preservation under strict privacy constraints.
*2) MedMNIST datasets:* These are evaluated across unbalanced non-IID, balanced non-IID, and IID splits. SENSE consistently matches centralized performance (Tables 3,12,11), even under high heterogeneity. Table 8 in Appendix, further shows low DE and high FS, confirming strong structural and similarity preservation. *3) Hyperbolic datasets (Airport, Amazon, DBLP):* For these datasets, the results in Table 2 highlight SENSE's geometry-aware design, achieving high FS and very low DE in non-Euclidean spaces. This confirms its adaptability across geometric regimes. Overall, SENSE effectively ensures:

Table 2: FS and DE for hyperbolic datasets in POINTWISE setting.

| Dataset | FS | DE |
|---|---|---|
| AIRPORT | 0.9992 | 0.000067 |
| AMAZON | 0.9945 | 0.00052 |
| DBLP | 0.9929 | 0.00073 |

- *Neighbor preservation:* High continuity and trustworthiness show SENSE keeps similar points close in the embedding, preserving semantics across clients.

Table 3: Performance of centralized (Van.) and SENSE variants under non-IID unbalanced splits.

| Data | Metric | t-SNE | | UMAP | | PHATE | | CNE(s=0) | | CNE(s=0.5) | | CNE(s=1) | |
|---|---|---|---|---|---|---|---|---|---|---|---|---|---|
| | | VAN. | SENSE | VAN. | SENSE | VAN. | SENSE | VAN. | SENSE | VAN. | SENSE | VAN. | SENSE |
| PneumoniaMNIST | Trust. | 0.9723 | 0.9712 | 0.7699 | 0.7673 | 0.8570 | 0.8590 | 0.9027 | 0.9008 | 0.8976 | 0.8952 | 0.8832 | 0.8806 |
| | Cont. | 0.9418 | 0.9383 | 0.9140 | 0.9154 | 0.9624 | 0.9608 | 0.9594 | 0.9591 | 0.9590 | 0.9583 | 0.9606 | 0.9599 |
| | Stead. | 0.7868 | 0.7932 | 0.6258 | 0.6168 | 0.7247 | 0.7204 | 0.7552 | 0.7591 | 0.7496 | 0.7461 | 0.7283 | 0.7341 |
| | Cohes. | 0.6991 | 0.6591 | 0.6318 | 0.6250 | 0.6953 | 0.6957 | 0.6983 | 0.7085 | 0.7052 | 0.7142 | 0.7015 | 0.7065 |
| BloodMNIST | Trust. | 0.9633 | 0.9609 | 0.8674 | 0.8632 | 0.8493 | 0.8513 | 0.8841 | 0.8816 | 0.8814 | 0.8795 | 0.8737 | 0.8715 |
| | Cont. | 0.9256 | 0.9375 | 0.9411 | 0.9401 | 0.9435 | 0.9428 | 0.9555 | 0.9552 | 0.9558 | 0.9556 | 0.9555 | 0.9552 |
| | Stead. | 0.7498 | 0.7480 | 0.6889 | 0.6874 | 0.6781 | 0.6851 | 0.7172 | 0.7323 | 0.7186 | 0.7216 | 0.7100 | 0.7132 |
| | Cohes. | 0.7242 | 0.7178 | 0.7253 | 0.7253 | 0.7456 | 0.7448 | 0.7462 | 0.7440 | 0.7384 | 0.7540 | 0.7533 | 0.7379 |
| BreastMNIST | Trust. | 0.9379 | 0.9378 | 0.7817 | 0.7998 | 0.8921 | 0.8884 | 0.9133 | 0.9117 | 0.9124 | 0.9113 | 0.9108 | 0.9108 |
| | Cont. | 0.9508 | 0.9481 | 0.8140 | 0.8247 | 0.9616 | 0.9563 | 0.9519 | 0.9515 | 0.9516 | 0.9513 | 0.9510 | 0.9509 |
| | Stead. | 0.8417 | 0.8329 | 0.5605 | 0.5550 | 0.8037 | 0.8149 | 0.8438 | 0.8480 | 0.8491 | 0.8495 | 0.8490 | 0.8398 |
| | Cohes. | 0.6091 | 0.6137 | 0.4095 | 0.4112 | 0.5668 | 0.5570 | 0.5777 | 0.5695 | 0.5807 | 0.5689 | 0.5675 | 0.5585 |
| DermaMNIST | Trust. | 0.9757 | 0.9770 | 0.7496 | 0.7466 | 0.8737 | 0.8728 | 0.9130 | 0.9121 | 0.9119 | 0.9116 | 0.9020 | 0.9021 |
| | Cont. | 0.9461 | 0.9572 | 0.9127 | 0.9122 | 0.9736 | 0.9730 | 0.9709 | 0.9713 | 0.9706 | 0.9707 | 0.9716 | 0.9715 |
| | Stead. | 0.7977 | 0.7979 | 0.5945 | 0.5936 | 0.7308 | 0.7319 | 0.7739 | 0.7689 | 0.7682 | 0.7686 | 0.7578 | 0.7553 |
| | Cohes. | 0.7147 | 0.7111 | 0.5586 | 0.5459 | 0.7127 | 0.7108 | 0.7268 | 0.7321 | 0.7385 | 0.7502 | 0.7438 | 0.7383 |
| RetinaMNIST | Trust. | 0.9797 | 0.9736 | 0.8793 | 0.8636 | 0.9161 | 0.9050 | 0.9486 | 0.9357 | 0.9475 | 0.9348 | 0.9451 | 0.9336 |
| | Cont. | 0.9496 | 0.9669 | 0.9273 | 0.9244 | 0.9738 | 0.9734 | 0.9720 | 0.9714 | 0.9707 | 0.9701 | 0.9678 | 0.9680 |
| | Stead. | 0.8442 | 0.8498 | 0.6307 | 0.5923 | 0.7559 | 0.7636 | 0.8267 | 0.8176 | 0.8196 | 0.8138 | 0.8158 | 0.8040 |
| | Cohes. | 0.6734 | 0.7281 | 0.5832 | 0.5828 | 0.6957 | 0.6991 | 0.7100 | 0.7137 | 0.7089 | 0.6982 | 0.6883 | 0.6990 |
| OrganCMNIST | Trust. | 0.9621 | 0.9387 | 0.8887 | 0.8867 | 0.8850 | 0.8871 | 0.9134 | 0.9041 | 0.9159 | 0.9056 | 0.9019 | 0.8907 |
| | Cont. | 0.9207 | 0.9170 | 0.9268 | 0.9247 | 0.9691 | 0.9699 | 0.9733 | 0.9693 | 0.9729 | 0.9685 | 0.9737 | 0.9696 |
| | Stead. | 0.7011 | 0.7855 | 0.7527 | 0.7718 | 0.7935 | 0.8093 | 0.8666 | 0.8755 | 0.8733 | 0.8722 | 0.8597 | 0.8607 |
| | Cohes. | 0.4685 | 0.5037 | 0.3322 | 0.3373 | 0.5431 | 0.5444 | 0.4653 | 0.5096 | 0.5681 | 0.5233 | 0.5745 | 0.5375 |
| OrganSMNIST | Trust. | 0.9552 | 0.9357 | 0.8741 | 0.8625 | 0.8792 | 0.8821 | 0.9114 | 0.9028 | 0.9126 | 0.9040 | 0.8993 | 0.8912 |
| | Cont. | 0.9214 | 0.9169 | 0.9246 | 0.9213 | 0.9684 | 0.9700 | 0.9738 | 0.9682 | 0.9731 | 0.9675 | 0.9736 | 0.9683 |
| | Stead. | 0.6765 | 0.7311 | 0.7222 | 0.7485 | 0.7809 | 0.7995 | 0.8609 | 0.8659 | 0.8664 | 0.8708 | 0.8561 | 0.8582 |
| | Cohes. | 0.4951 | 0.4814 | 0.3603 | 0.3211 | 0.5198 | 0.5343 | 0.4704 | 0.44009 | 0.5192 | 0.4833 | 0.5155 | 0.5033 |
| german-credit | Trust. | 0.9745 | 0.9543 | 0.9514 | 0.9294 | 0.8555 | 0.8394 | 0.9337 | 0.9124 | 0.9380 | 0.9072 | 0.9336 | 0.9092 |
| | Cont. | 0.9583 | 0.9424 | 0.9604 | 0.9410 | 0.9481 | 0.9255 | 0.9571 | 0.9438 | 0.9576 | 0.9438 | 0.9571 | 0.9440 |
| | Stead. | 0.8576 | 0.8248 | 0.8313 | 0.7933 | 0.7483 | 0.7061 | 0.8398 | 0.7921 | 0.8479 | 0.7855 | 0.8436 | 0.7906 |
| | Cohes. | 0.6774 | 0.6755 | 0.6638 | 0.6568 | 0.6893 | 0.6745 | 0.6446 | 0.6551 | 0.6575 | 0.6481 | 0.6513 | 0.6676 |

- *Similarity recovery:* Despite no raw data access, SENSE accurately approximates pairwise distances evidenced by low DE and high FS.
- *Cluster structure:* Comparable steadiness and cohesiveness confirm that SENSE maintains cluster alignment without fragmentation.

**Visualization.** Figure 2 shows global embeddings learned by SENSE on MNIST in the MULTISITE setting with 25,000 NA samples across 10 clients in an unbalanced non-IID split. Using only 783 anchors ($d_h - 1$), SENSE constructs high-quality embeddings without accessing or sharing raw features. Embeddings from t-SNE, UMAP, PHATE, and CNE cleanly separate semantic groups, preserving local neighborhoods and global cluster topology. By estimating inter-client similarities, SENSE enables meaningful inter-client positive/negative contrastive pairs. This highlights its ability to learn structure-preserving, privacy-compliant embeddings in decentralized, heterogeneous settings. Additional visualizations are in the Appendix A.14.

### 4.3 ABLATION STUDY.

To validate Theorems 3.1, 3.2, and Lemma 1, we perform an ablation study by varying anchor count from $d_h - \epsilon$ to $d_h + \epsilon$. We evaluate SENSE using five normalized metrics, plotted in Figure 3: (i) *Cosine Similarity* (Nguyen & Bai, 2010) between ground-truth $X'_{\text{NA}}$ and reconstructed latent embeddings $\widehat{X}_{\text{NA}}$; (ii) *Distance Error* and (iii) *F-score* (Sec. 4.1); (iv) *Pearson Correlation* ($\rho$) (Sedgwick, 2012) over NA–NA distances; and (v) *Frobenius Norm Error* ($X_{\text{frob}}$) (Kannan, 1989), capturing reconstruction loss (full definitions in Appendix A.15). Key observations from the study:

- *Effective with few anchors:* Even with anchor count well below $d_h$ (e.g., $d_h - 100$), SENSE achieves high F-score, low distance error, and strong cosine similarity, showing robust neighborhood preservation in resource-constrained settings.
- *Privacy-compliant reconstruction:* As anchors approach $d_h$, cosine and Pearson scores improve. Beyond $d_h + 1$, near-zero Frobenius error indicates possible exact recovery highlighting the need to limit anchor count to preserve privacy.
- *Structural consistency:* Pearson correlation rises with anchor count, saturating near 1.0 at $d_h + 1$, with corresponding drops in Frobenius error confirming theoretical bounds for exact recovery.
- *Metric alignment with theoretical thresholds:* Across datasets, all metrics converge near $d_h$, with diminishing gains beyond matching theoretical thresholds.

These results validate that SENSE achieves high-fidelity, privacy-compliant reconstruction with minimal anchors, making it scalable and effective in decentralized settings with limited observability.

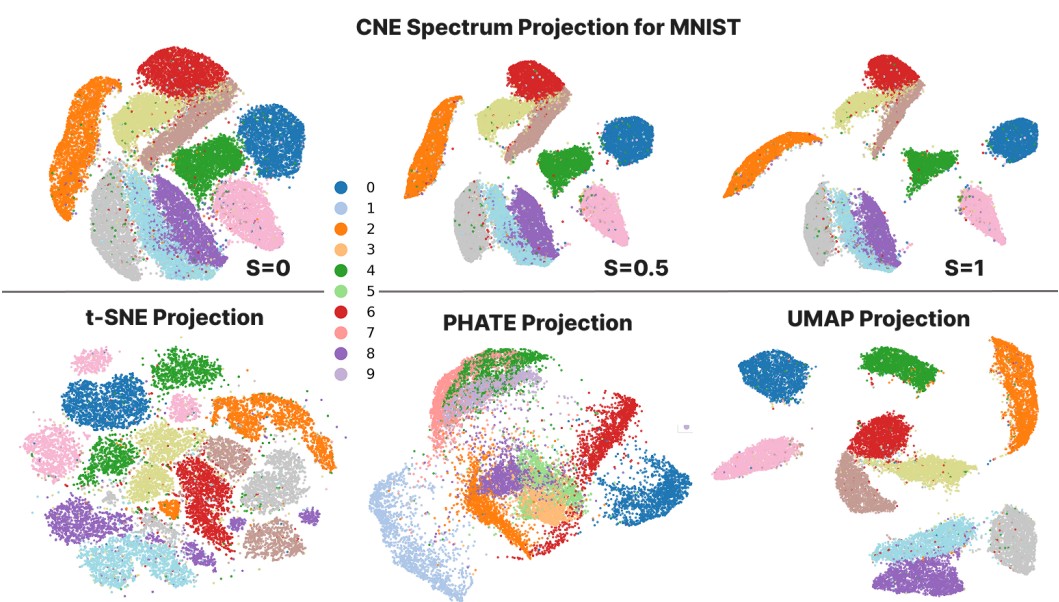

Figure 2: Global embeddings of MNIST under the MULTISITE setting. **Top:** CNE spectrum with SENSE. **Bottom:** t-SNE, PHATE, and UMAP embeddings generated via SENSE without any raw feature sharing. All embeddings preserve global structure while ensuring privacy.

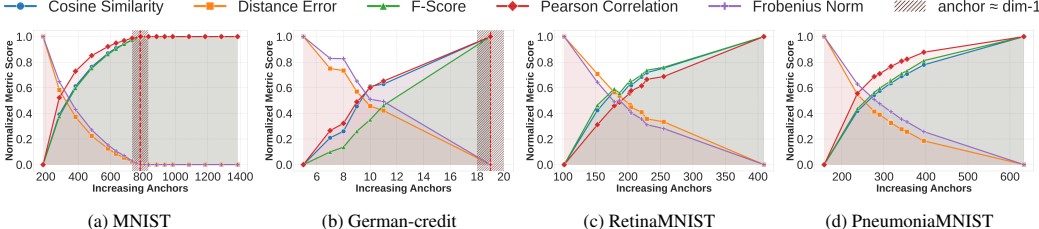

Figure 3: Impact of anchor count on normalized metric scores under non-IID unbalanced distributions. The red vertical line denotes the theoretical privacy threshold at $d_h - 1$ (783 for MNIST, 19 for German Credit), beyond which exact recovery may be possible. For Retina and Pneumonia, this threshold lies outside the x-axis range, resulting in monotonic performance gains. Trends confirm trade-offs between reconstruction fidelity and privacy risk as anchor count increases.

### 4.4 DISCUSSION.

Due to space constraints, we provide the extended discussion of SENSE and additional empirical results in Appendix A.16. It addresses four key aspects: (i) how SENSE adapts to dynamic evolving distributed environments via out-of-sample embedding; (ii) scalability and runtime on large-scale datasets such as Tiny ImageNet (Le & Yang, 2015) and SVHN (Netzer et al., 2011) (Table 13), with a detailed breakdown of pipeline stages and complexity analysis in Table 14; (iii) a curated privacy example showing that embeddings preserve structural relationships while preventing recovery of sensitive attributes (Table 15); and (iv) why SENSE avoids noise-based privacy (Table 16 and 17), since injected noise quickly degrades embedding quality, whereas geometric underdetermination preserves both fidelity and privacy.

## 5 CONCLUSION

We proposed SENSE, a geometry-aware framework for decentralized representation learning that reconstructs global geometry from sparse anchor-based distances, enabling projections without raw data exchange. By combining structured matrix completion with classical DR methods, SENSE supports both Euclidean and hyperbolic spaces and adapts to multiple deployment settings. Experiments show that SENSE is effective even with few anchors, achieving strong neighborhood and cluster preservation while matching centralized baselines under strict privacy and communication constraints. These results highlight SENSE as a scalable, privacy-preserving solution for collaborative representation learning in heterogeneous, non-IID environments.

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

# A  APPENDIX

## A.1  NEIGHBOR EMBEDDING (NE).

**Definition A.1** *t-SNE models $p_{ij}$ as symmetrized conditional probabilities using Gaussian kernels:*
$p_{j|i} \propto \exp(-\|x_i - x_j\|^2/2\sigma_i^2)$, *with* $p_{ij} = \frac{p_{j|i}+p_{i|j}}{2n}$. *Low-dimensional similarities are computed using a heavy-tailed Student-t kernel:* $q_{ij} \propto (1 + \|y_i - y_j\|^2)^{-1}$. *The loss minimizes the KL divergence:*

$$\mathcal{L}_{tSNE} = \sum_{i \neq j} p_{ij} \log \frac{p_{ij}}{q_{ij}}.$$

**Definition A.2** *UMAP defines $p_{j|i} = \exp(-(\|x_i - x_j\| - \rho_i)/\tau_i)$ using adaptive exponential kernels, where $\rho_i$ is the local connectivity threshold. Symmetrized $p_{ij}$ is computed via fuzzy set union. In the embedding space, $q_{ij} = (1 + a\|y_i - y_j\|^2)^{-b}$ with fixed parameters $(a, b)$. The loss is a weighted binary cross-entropy:*

$$\mathcal{L}_{UMAP} = \sum_{i \neq j} \left[ p_{ij} \log \frac{p_{ij}}{q_{ij}} + (1 - p_{ij}) \log \frac{1 - p_{ij}}{1 - q_{ij}} \right].$$

## A.2  CONTRASTIVE NEIGHBOR EMBEDDING (CNE).

**Definition A.3** *Given a kNN graph, high-dimensional similarities are binary: $S_{ij}^{d_h} = 1$ if $x_j \in kNN(x_i)$, and $0$ otherwise. In the embedding space, similarities are defined using a Cauchy kernel: $S_{ij}^{d_l} = \phi(\mathbf{y}_i, \mathbf{y}_j) = \frac{1}{1+\|\mathbf{y}_i - \mathbf{y}_j\|^2}$. The CNE objective combines attractive and repulsive forces:*

$$\mathcal{L}(\theta) = -\mathbb{E}_{(i,j) \sim p_i} \log \phi(f_\theta(\mathbf{x}_i), f_\theta(\mathbf{x}_j)) - b\mathbb{E}_{(i,j)} \log(1 - \phi(f_\theta(\mathbf{x}_i), f_\theta(\mathbf{x}_j))),$$

*where $p_i$ samples positive pairs and $b > 0$ balances the repulsion term.*

## A.3  HYPERBOLIC MODELS AND DISTANCE CALCULATION.

There are several equivalent models of hyperbolic geometry exist, including the Poincaré ball model, lorentz model (or hyperboloid model) and the upper half-space model. The mathematical framework of the $d$-dimensional hyperboloid model of hyperbolic geometry is deined as follows:

For $x, y \in \mathbb{R}^{d+1}$, the Lorentz product is an indefinite inner product given by,

$$x \circ y := x_1 y_1 - (x_2 y_2 + \cdots + x_{d+1} y_{d+1}). \tag{12}$$

The real vector space $\mathbb{R}^{d+1}$ equipped with this inner product is called *Lorentz space*, denoted by $\mathbb{R}^{1,d}$. It contains the *positive Lorentz space* as a subset:

$$\mathbb{R}_+^{1,d} := \left\{ x \in \mathbb{R}^{1,d} \; : \; x_1 > 0 \right\}.$$

Within $\mathbb{R}_+^{1,d}$, the *single-sheet hyperboloid* $\mathbb{H}^{d_h}$ is given by

$$\mathbb{H}^{d_h} := \left\{ x \in \mathbb{R}^{1,d} \; : \; x \circ x = 1, \; x_1 > 0 \right\}. \tag{13}$$

The hyperboloid model in dimension $d$ with curvature $-\kappa$ (for $\kappa > 0$) consists of $\mathbb{H}^{d_h}$ endowed with the hyperbolic distance:

$$d_{\mathbb{H}}^\kappa(x, y) = \frac{1}{\sqrt{\kappa}} \text{arcosh}(x \circ y), \quad x, y \in \mathbb{H}^{d_h}. \tag{14}$$

The distance $d_{\mathbb{H}}^\kappa$ is a valid metric on $\mathbb{H}^{d_h}$, it is positive definite and satisfies the triangle inequality. Moreover, equipped with the metric tensor:

$$ds^2 = \frac{1}{\kappa}(dx \circ dx),$$

the hyperboloid $\mathbb{H}^{d_h}$ becomes a Riemannian manifold of constant sectional curvature $-\kappa$, and $d_{\mathbb{H}}^\kappa$ corresponds exactly to its geodesic distance. In particular, the curvature $\kappa$ does not alter the definition of the manifold $\mathbb{H}^{d_h}$ itself, but only scales the distance metric. Just as Euclidean space is the canonical model for zero curvature, hyperbolic space is the canonical geometry for constant negative curvature.

### A.3.1 POINCARÉ BALL MODEL.

The Poincaré ball model is the most widely used formulation of hyperbolic space in machine learning (Nickel & Kiela, 2017; Ganea et al., 2018). It defines the $n$-dimensional hyperbolic space as $\mathbb{B}^n = \{x \in \mathbb{R}^n : \|x\| < 1\}$ with Riemannian metric $g_x = \left(\frac{2}{1-\|x\|^2}\right)^2 I_n$. The hyperbolic distance between two points $u, v \in \mathbb{B}^n$ is:

$$d_{\mathbb{B}^n}(u, v) = \operatorname{arcosh}\left(1 + \frac{2\|u - v\|^2}{(1 - \|u\|^2)(1 - \|v\|^2)}\right). \tag{15}$$

This distance increases exponentially near the boundary, enabling natural hierarchical embeddings where central points correspond to root nodes and peripheral points to leaves.

### A.4 CO-SNE

**Definition A.4** *CO-SNE defines the similarities via hyperbolic normal kernels in the high-dimensional Poincaré ball $\mathbb{B}^n$: $p_{j|i} = \exp\left(-d_{\mathbb{B}^n}(x_i, x_j)^2/2\sigma_i^2\right)/Z_i$, with $p_{ij} = (p_{j|i} + p_{i|j})/2m$. In the embedding space $\mathbb{B}^2$, similarities use a hyperbolic Cauchy kernel: $q_{ij} = \gamma^2/(d_{\mathbb{B}^2}(y_i, y_j)^2 + \gamma^2)/Z$. The loss combines KL divergence with a norm-based regularizer:*

$$\mathcal{L}_{\text{CO-SNE}} = \lambda_1 \sum_{i,j} p_{ij} \log \frac{p_{ij}}{q_{ij}} + \lambda_2 \sum_i (\|x_i\|^2 - \|y_i\|^2)^2. \tag{16}$$

### A.5 CLASSICAL MDS

Utilizing the measurements of distances among pairs of objects, MDS (multidimensional scaling) finds a representation of each object in $d$ - dimensional space such that the distances are preserved in the estimated configuration as closely as possible. To validate the goodness-of-fit measure, MDS optimizes the loss function (known as "Stress"($\sigma$)) given by:

$$\sigma(X) = \min_X \sum_{i<j \leq N} w_{ij} \left(\delta_{ij} - d_{ij}(X)\right)^2, \tag{17}$$

, where the observation mask is $W$ where $w_{ij} = 1$ if the distance $\delta_{ij}$ is known and $w_{ij} = 0$ otherwise, with the block structure:

$$W = \begin{bmatrix} \mathbf{0}_{N \times N} & \mathbf{1}_{N \times M} \\ \mathbf{1}_{M \times N}^\top & \mathbf{1}_{M \times M} \end{bmatrix} \tag{18}$$

where $\mathbf{0}$ and $\mathbf{1}$ denote matrices of zeros and ones, respectively and $X$ represents the computed configuration, $d_{ij}(X) = \|x_i - x_j\|$ is the Euclidean distance between nodes $i$ and $j$, $\delta_{ij}$ is the measured distance computed privately. Placing the weights of unknown inter-user distance to zero, the weight matrix $W$ can be partitioned into block matrices as shown in 18, where $11_{N,M}$ is a matrix of ones with shape $N \times M$. De Leeuw (De Leeuw, 2005) applied an iterative method called SMACOF (Scaling by Majorizing a Convex Function) to estimate the configuration $X$. As the objective is a non-convex function, SMACOF minimizes the stress using the simple quadratic function $\tau(X, Z)$ which bounds $\sigma(X)$ (the complicated function) from above and meets the surface at the so-called supporting point $Z$ as defined below:

$$\sigma(X) \leq \tau(X, Z) = \sum_{i<j} w_{ij} \delta_{ij}^2 + \sum_{i<j} w_{ij} d_{ij}^2(X) - 2 \sum_{i<j} w_{ij} \delta_{ij}^2 \frac{(x_i - x_j)^T (z_i - z_j)}{\|z_i - z_j\|} \tag{19}$$

Equation 19 can be written in matrix form as:

$$\tau(X, Z) = C + \operatorname{tr}\left(X^T V X\right) - 2 \operatorname{tr}\left(X^T B(Z) Z\right). \tag{20}$$

The iterative solution which guarantees monotone convergence of stress (De Leeuw, 1988) is given by equation 21, where $Z = X^{k-1}$:

$$X^{(k)} = \min_X \tau(X, Z) = V^\dagger B(X^{(k-1)}) X^{(k-1)} \tag{21}$$

This algorithm offers flexibility to embed features in any dimension other than $d$, which enables the handling of high-dimensional data and also meets privacy constraints. As $V$ is not of full rank, hence

the Moore-Penrose pseudoinverse $V^\dagger$ is used. The elements of the matrix $B(X)$ and $V$ are defined in equation 22.

$$
b_{ij} = \begin{cases} -\dfrac{w_{ij}\delta_{ij}}{d_{ij}(\mathbf{X})}, & \text{if } d_{ij}(\mathbf{X}) \neq 0, \ i \neq j \\ 0, & \text{if } d_{ij}(\mathbf{X}) = 0, \ i \neq j \\ -\sum\limits_{j=1, \ j\neq i}^{N} b_{ij}, & \text{if } i = j \end{cases}
\tag{22}
$$

$$
v_{ij} = \begin{cases} -w_{ij}, & \text{if } i \neq j \\ -\sum\limits_{j=1, \ j\neq i}^{N} v_{ij}, & \text{if } i = j \end{cases}
$$

## A.6 SENSE: PSEUDOCODE

---
**Algorithm 1** SENSE Framework

---
**Require:** Anchors $\mathbf{X}_A \in \mathbb{R}^{K \times d_{\mathrm{h}}}$, client datasets $\{\mathcal{D}_m = \{x_i^m\}_{i=1}^{N_m}\}_{m=1}^{M}$, target dim $d_\ell$, high/low geometry $\mathbb{G}_{\mathrm{high}} \in \{\mathbb{R}^{d_h}, \mathbb{H}^{d_h}\}$, $\mathbb{G}_{\mathrm{low}} \in \{\mathbb{R}^{d_\ell}, \mathbb{H}^{d_\ell}\}$
**Ensure:** Global embeddings $\{\mathbf{Y}^m \in \mathbb{G}_{\mathrm{low}}^{N_m}\}_{m=1}^{M}$
1: Server broadcasts $\mathbf{X}_A$ to all clients
2: **for** each client $\mathcal{C}_m$ **do**
3:    Compute distances $\mathbf{d}_i^m = \mathcal{D}_{\mathbb{G}_{\mathrm{high}}}(x_i^m, \mathbf{X}_A)$ for all $x_i^m \in \mathcal{D}_m$
4:    Send $\{\mathbf{d}_i^m\}_{i=1}^{N_m}$ to server
5: **end for**
6: Server builds observed matrix $\mathbf{D}_\Omega$ using $E$, $F$, (optionally $G$)
7: Complete $\widehat{\mathbf{D}}$ via structured matrix completion; extract $\widehat{\mathbf{G}}$
8: Compute similarities $S^{d_{\mathrm{h}}}$ from $\widehat{\mathbf{G}}$ using kernel $f$ (see Eqns 6, 7)
9: Learn embedding $\mathbf{Y}$ in $\mathbb{G}_{\mathrm{low}}$ using NE, contrastive, or CO-SNE objective

---

## A.7 SENSE VIA ANCHORED-MDS: PSEUDOCODE

---
**Algorithm 2** SENSE via Anchored-MDS

---
**Require:** Anchor embeddings $X_A \in \mathbb{R}^{K \times d_h}$, observed entries $\mathcal{P}_\Omega(D)$, target dim $d_h$, tolerance $\epsilon$, max iterations $T$
**Ensure:** Reconstructed embeddings $X_{NA} \in \mathbb{R}^{N \times d_h}$
1: Initialize $X_{NA}^{(0)}$ randomly, set $k \leftarrow 1$
2: **while** $k \leq T$ **do**
3:    Form $X^{(k-1)} = \begin{bmatrix} X_A & X_{NA}^{(k-1)} \end{bmatrix}^T$
4:    Compute $\mathcal{P}_\Omega(D(X^{(k-1)}))$
5:    Construct $W$ and compute $V$, $B(X^{(k-1)})$ respecting $\Omega$
6:    Update $X_{NA}^{(k)}$ using Eq. 11
7:    If stress improvement $< \epsilon$, **break**; else $k \leftarrow k+1$
8: **end while**
9: **return** $X_{NA}^{(k)}$

---

## A.8 ANCHOR GENERATION

In the proposed method, distribution of the anchor data is critical. These anchors are not private; they act as public or semi-public landmarks, akin to those in GPS (Shang & Ruml, 2004) or radar systems (Iannucci et al., 2020). This is standard in localization literature (Di Franco et al., 2017; Khan et al., 2009), where landmarks aid positioning but are not privacy-sensitive (Koledoye et al., 2017). Our privacy definition protects only the high-dimensional features of NA points. Anchors are fixed, visible, and either synthetic, public, or explicitly consented. Not part of any client's private data. We do not rely on anchor secrecy but instead limit their quantity to preserve ambiguity.

The anchor is a common information shared between all the clients. The anchor data is generated randomly or by open data for securing privacy. The proper scheduling of the anchors has a significant

impact on the overall performance and accuracy of the framework. There are several factors to consider when developing the anchor scheduling strategy, including:

**1) Number of anchors** : The number of anchors used in the framework has a direct impact on the algorithmic performance. Too few anchors may not preserve the structural information while ensuring privacy, while too many anchors may violate privacy.

**Choice of $K$:** We sample anchors such that the total (global + local) anchors satisfy $K = d_h - 1$, where $d_h$ is the input dimension. This is not arbitrary; it is *theoretically optimal* under our privacy guarantee. By Theorem 3.1, if $K < d_h$, privacy is ensured. Thus, $K = d_h - 1$ is the largest safe choice that preserves privacy while still yielding strong approximations. There is an inherent privacy-utility-compute trade-off:

1. Higher $K$: better fidelity, higher runtime, weaker privacy.
2. Lower $K$: stronger privacy, lower compute cost, possible fidelity drop.

We empirically validate this trade-off on a synthetic dataset where we took $K$ anchors and $N$ NA points in a pointwise setting with $d_h$ dimension.

**For low-dimensional data** ($d_h = 100$, $N = 500$):

- $K = 99 \Rightarrow$ FS = 0.79, time = 16s
- $K = 90 \Rightarrow$ FS = 0.78, time = 13s

**For high-dimensional data** ($d_h = 700$, $N = 500$):

- $K = 699 \Rightarrow$ FS = 0.82, time = 660s
- $K = 350 \Rightarrow$ FS = 0.78, time = 312s

Thus, for high-dimensional settings where $d_h \gg N$, using fewer anchors (e.g., $K < d_h - 1$) yields substantial computational gains while preserving utility. This is particularly useful in large-scale deployments such as hospital networks, financial networks, social media platforms and IOT networks.

**2) Anchor Geometry:** Beyond count, anchor geometry also impacts fidelity. We show that affinely independent anchors improve reconstruction quality. On the DIGITS dataset ($N = 1797$, $d_h = 64$) (Alpaydin & Kaynak, 1998), we fixed $K = 63$ and varied the anchor matrix rank $r$. The non-anchor set consisted of 1000 points, split across 10 clients (multisite-full). Table 4 validates that *higher affine rank of anchors improves neighborhood fidelity*, consistent with our theory.

Table 4: Effect of anchor matrix rank $r$ on FS and DE in the DIGITS dataset ($K = 63$). Higher rank improves fidelity.

| $r$ | FS | DE |
| --- | --- | --- |
| 10 | 0.524 | 0.0829 |
| 20 | 0.6242 | 0.0604 |
| 30 | 0.716 | 0.0479 |
| 40 | 0.7955 | 0.0361 |
| 50 | 0.8315 | 0.0311 |
| 60 | 0.858 | 0.0259 |
| 63 | 0.861 | 0.0263 |

**3) Selection criteria** : The criteria used to select anchors can also impact the performance of the system. Selecting anchors from the same probability distribution as of the underlying user data may be more effective than selecting them at random. For example, the data distribution of patient similarity networks or social networks will depend on factors including a number of patients/users or similarity of patients/connection between users. We also empirically study how anchor selection affects downstream performance. Random anchor sampling performs worse than anchors sampled from the underlying data distribution. Table 5 confirms that *in-distribution anchors preserve structure better*. Setup: 10 clients, 1000 non-anchor points, multisite-full setting.

**4) Practical Anchor Sources.** In practical deployments, anchors are selected based on domain knowledge and are not sampled arbitrarily from private data. Eg, Healthcare: Publicly released

Table 5: Effect of anchor selection on fidelity (FS) and distance error (DE). In-distribution anchors outperform random anchors across datasets.

| Data | Anchor-Type | $K$ | FS | DE |
|------|-------------|-----|-----|-----|
| Digits | In-Distri | 60 | 0.900 | 0.027 |
| | Rand | 60 | 0.345 | 0.382 |
| MNIST | In-Distri | 783 | 0.967 | 0.006 |
| | Rand | 783 | 0.170 | 0.602 |
| BloodMNIST | In-Distri | 2351 | 0.961 | 0.005 |
| | Rand | 2351 | 0.176 | 0.892 |

reference scans or patient-consented samples (Johnson et al., 2016). Finance: Standard transaction patterns or aggregated customer profiles (Awosika et al., 2024b). Genomics: Population-level reference genomes (e.g., 1000 Genomes, UK Biobank) (Bycroft et al., 2018; Regev et al., 2017). Synthetic Anchors: Via MMD minimization (Qiao et al., 2024), though limited in coverage and potentially adversarial. Trusted server-curated anchors are auditable, robust, and independent of client records, reducing leakage risks like membership inference (Shokri et al., 2017). This design is further supported by theory on low-rank recovery via anchor distances (Lichtenberg & Tasissa, 2024b), matrix completion in non-orthogonal bases (Tasissa & Lai, 2019), and Gram matrix-based localization (Mishra et al., 2011).

Table 6: Observed index sets $\Omega$ used for SENSE under each client configuration. Here, $\mathcal{A}_G$ denotes global anchors, $\mathcal{A}_L^{(j)}$ are local anchors accessible only to client $j$, and $\mathcal{X}^{(m)}$ are NA indices at client $m$. Binary masks $W_F$ and $W_G$ indicate anchor-to-NA and intra-client NA–NA visibility. Observed distances are used to construct $V$, $B(X)$, and select relevant rows of $X_A$ for embedding computation.

| SENSE Setting | Observed Index Set $\Omega$ |
|---------------|------------------------------|
| **Pointwise-Full** | Each client holds one NA. All anchor-to-NA distances are known; no NA–NA or local anchor information. $\Omega = \{(i,j) : i \in \mathcal{A}_G, \ j \in [K+1, K+N]\} \cup \{(j,i) : i \in \mathcal{A}_G, \ j \in [K+1, K+N]\}$ |
| **Pointwise-Partial** | Each client holds one NA. Global anchors $\mathcal{A}_G$ are shared across all clients. Local anchors $\mathcal{A}_L^{(j)}$ are only accessible to client $j$. $\Omega = \bigcup_{j=1}^{N} \left( (\mathcal{A}_G \cup \mathcal{A}_L^{(j)}) \times \{K+j\} \cup \{K+j\} \times (\mathcal{A}_G \cup \mathcal{A}_L^{(j)}) \right)$ |
| **Multisite-Full** | Each client holds multiple NAs. All anchor-to-NA distances are known. Intra-client NA–NA distances are observed. $\Omega = \{(i,j) : i \in \mathcal{A}_G, j \in [K+1, K+N]\} \cup \{(j,i) : i \in \mathcal{A}_G, j \in [K+1, K+N]\} \cup \bigcup_{m=1}^{M}(\mathcal{X}^{(m)} \times \mathcal{X}^{(m)})$ |
| **Multisite-Partial** | Each client holds multiple NAs. Anchor-to-NA distances are partially known via $W_F$ (global + local anchors). Intra-client NA–NA distances are observed via $W_G$. $\Omega = \{(i, j+K) : W_F[i,j] = 1\} \cup \{(j+K, i) : W_F[i,j] = 1\} \cup \{(i,j) : W_G[i,j] = 1\}$ |

A.9 THEORETICAL PROOFS.

Unlike some EDG (Tasissa & Lai, 2019) methods that assume uniform random sampling of pairwise distances, SENSE uses a structured sampling scheme where anchor-to-NA distances are measured by design. This enables deterministic recovery guarantees based on geometric conditions (e.g., connectivity to affinely independent anchors), avoiding reliance on probabilistic bounds from random sampling.

**Proof A.1** *Consider a network in $d_h$-dimensional Euclidean space $\mathbb{R}^{d_h}$, comprising anchors $A = \{A_1, A_2, \ldots, A_K\}$ and non-anchor nodes $P = \{P_1, P_2, \ldots, P_N\}$, with feature vectors $\boldsymbol{x_i} \in \mathbb{R}^{d_h}$. Anchors locations are known, while non-anchors need estimation. Previous work (Khan et al., 2009) shows that in $\mathbb{R}^{d_h}$, a minimum of $(d+1)$ anchors with known locations is required to locate $N$ non-anchor nodes. The utilization of anchors for distributed sensor localization constitutes a thoroughly investigated domain, underpinned by the following assumptions:*

- **(A1)** *Non-anchor nodes lie inside the convex hull of the anchors, i.e., $C(P) \subseteq C(A)$.*
- **(A2)** *Each non-anchor node $P_i$ has at least one set of neighbor nodes $N_i \subset (A \cup P)$ with $|N_i| = d_h + 1$ such that $i$ lies inside $C(N_i)$.*
- **(A3)** *In the set $\{i \cup N_i\}$, every non-anchor node $i$ can obtain the inter-node distances among all nodes.*

*However, to accurately recover features in $\mathbb{R}^{d_h}$, at least $d_h$ anchors are necessary, even if non-anchors are placed in any location. Thus, having fewer than $d_h$ anchors, i.e., $K < d_h$, guarantees that exact feature embeddings cannot be obtained, ensuring privacy.*

**Proof A.2** *Each NA point $\boldsymbol{x}_j \in \mathbb{R}^{d_h}$ computes squared distances to a subset of anchors indexed by $\mathcal{I}_j$, with $r_j = |\mathcal{I}_j|$. This yields $r_j$ quadratic constraints of the form:*

$$\|\boldsymbol{x}_j - \boldsymbol{a}_i\|^2 = d_{hij}^2, \quad \forall i \in \mathcal{I}_j.$$

*To analyze identifiability, fix a reference anchor $\boldsymbol{a}_k \in \mathcal{I}_G$ from the global anchor set, and consider the difference of equations relative to this reference:*

$$\|\boldsymbol{x}_j - \boldsymbol{a}_i\|^2 - \|\boldsymbol{x}_j - \boldsymbol{a}_k\|^2 = d_{hij}^2 - d_{hkj}^2.$$

*Expanding and simplifying yields the linear system:*

$$2(\boldsymbol{a}_k - \boldsymbol{a}_i)^\top \boldsymbol{x}_j = \|\boldsymbol{a}_k\|^2 - \|\boldsymbol{a}_i\|^2 + d_{hij}^2 - d_{hkj}^2, \quad \forall i \in \mathcal{I}_j \setminus \{k\}.$$

*Letting $A_j \in \mathbb{R}^{(r_j-1) \times d}$ denote the coefficient matrix and $\boldsymbol{b}_j$ the RHS vector, we write:*

$$A_j \boldsymbol{x}_j = \boldsymbol{b}_j.$$

*This is a system of $r_j - 1$ linear equations in $d_h$ unknowns. If $r_j < d_h + 1$, then $\text{rank}(A_j) \leq r_j - 1 < d_h$, and the solution set $\{\boldsymbol{x}_j \in \mathbb{R}^{d_h} : A_j \boldsymbol{x}_j = \boldsymbol{b}_j\}$ forms an affine subspace of dimension at least $d_h - r_j + 1$. Hence, infinitely many solutions exist that satisfy the same anchor distances, preventing exact recovery of $\boldsymbol{x}_j$.*

*To ensure privacy across all clients (both pointwise and multisite), we enforce:*

$$|\mathcal{I}_j| = K_G + K_L^{(j)} \leq d_h, \quad \forall j \in [N],$$

*where $K_L^{(j)}$ is the number of local anchors accessible to $\boldsymbol{x}_j$. In the multisite case, local anchors are restricted to the corresponding client, and global anchors are common across all clients. This structure ensures that even with partial anchor visibility, each client's feature vector cannot be uniquely recovered from its observed distances.*

**Remark 3** *Each anchor distance imposes a quadratic constraint on the unknown $\boldsymbol{x}_j \in \mathbb{R}^{d_h}$. If the number of constraints $r_j$ is less than the ambient dimension $d$, the system is underdetermined and has infinitely many solutions. Thus, SENSE preserves privacy by bounding the number of anchor distances accessible to each client.*

**Proof A.3** *From Theorem 3.1 (Exact Recovery) in (Keller-Ressel & Nargang, 2022), the L-HYDRA algorithm guarantees recovery up to isometry only if $K \geq d_h$ and the $K$ anchors are in general position (not lying on a single hyperbolic hyperplane). If $K < d_h$, then the system of equations defined by $E$ and $F$ is underdetermined: the landmarks do not span $\mathbb{H}_h^{d_h}$, and multiple embeddings of the NA points are consistent with the observed distances. Hence, SENSE ensures privacy by choosing $K < d_h$, preventing unique reconstruction of private client embeddings.*

## A.10 WHY $K < d_h$ AND NOT $K \leq d_h$?

While both $k < d$ and $k = d$ can contribute to privacy, our choice of $k < d$ is a deliberate design decision driven by the well-known mirror ambiguity problem (details in NOTE2) inherent in certain geometric transformations when $k = d$.

*Why Choosing $K < d$ Anchors is Better for Privacy than $K = d$:* In distance-based reconstruction problems such as the Distance Geometry Problem (DGP) (Liberti et al., 2014), network localization (Lichtenberg & Tasissa, 2024b), or hyperbolic embedding recovery (Keller-Ressel & Nargang, 2022), the number and configuration of anchor points (i.e., known reference points with distance access to

an unknown point) directly affect how precisely an adversary can reconstruct the hidden point. From a privacy perspective, we aim to make the reconstruction problem ambiguous, so that the true point remains hidden among many plausible candidates. We argue that choosing $K = D - 1$ (or fewer) anchors is preferable to choosing $K = D$, especially when protecting sensitive features such as location or identity (in case of hospital patients or other critical attributes). This relates to ambiguity, recoverability, and robustness under both Euclidean and hyperbolic regimes.

**Case 1:** $K = D$, **Affinely Independent Anchors** → **Small, Structured Ambiguity:** When an adversary knows distances from an unknown point $x \in \mathbb{R}^D$ to $D$ affinely independent anchors, the solution set for $x$ becomes tightly constrained: The point lies on a 1D manifold, typically a circle (in 2D/3D). This is the intersection of $D$ hyperspheres in $\mathbb{R}^D$, reducing degrees of freedom to 1 (rotation around the affine hull of the anchors). *Example:* In 2D: 2 non-collinear anchors ⇒ two symmetric positions across the anchor line. In 3D: 3 non-coplanar anchors ⇒ solution lies on a circle (Liberti et al., 2014) and also in (Liberti et al., 2014; Biswas & Ye, 2004; Fang, 1986). This is called *structured ambiguity*: the solution isn't unique, but the adversary can narrow it down to a small, reversible set, which weakens privacy. *BUT: $K = D$ Affinely Dependent Anchors → Degeneracy Risk:* If the $D$ anchors are affinely dependent (e.g., lie on a hyperplane), the problem becomes ill-posed. The linear system degenerates, and the solution set can inflate from a curve to a surface or even higher. *Example:* In 2D: 2 collinear anchors ⇒ infinite feasible points on a circle. In 3D: 3 coplanar anchors ⇒ solution lies on a cylinder surface (2D ambiguity) (Liberti et al., 2014). Thus, while ambiguity helps privacy, this case is fragile and depends on affine dependence, which is difficult to control/detect in high dimensions.

**Case 2:** $K = D - 1$, **Affinely Independent Anchors** → **Robust, Structured Ambiguity:** When the number of anchors is reduced to $K = D - 1$ and they are affinely independent: The intersection of $D - 1$ hyperspheres in $\mathbb{R}^D$ leaves the unknown point on a 1D manifold (if $K = D - 1$) or a 2D manifold (if $K = D - 2$). This increases ambiguity while preserving structure and analyzability. *Example:* In 3D: 2 non-collinear anchors ⇒ solution lies on a sphere surface (2D ambiguity). In 4D: 3 anchors ⇒ solution lies on a 2D manifold in $\mathbb{R}^4$ (Liberti et al., 2014). *This ambiguity is independent of affine structure:* even with poorly placed anchors, ambiguity remains large enough to preserve privacy.

**Case 3: Moreover, L-HYDRA Shows,** $K \geq d$ **Enables Exact Recovery** → **Privacy Violation:** In hyperbolic space, the L-HYDRA framework (Keller-Ressel & Nargang, 2022) (Theorem 3.1) shows that if $l \geq d$ well-placed landmarks are known, exact recovery up to isometry is possible. While useful for learning, this completely breaks privacy at $K = d$. Thus, in the SENSE framework, we limit the number of anchor constraints to prevent identifiability, ensure privacy, and make it more generic and applicable across all cases.

Why $K = D - 1$ is the safe design choice for privacy? We prefer using $K = D - 1$ anchors because:

1. *More Ambiguity = More Privacy:* Reducing constraints expands the feasible set (e.g., from a 1D curve to a 2D surface). This makes inference harder for an adversary.
2. *Independent of Affine Structure:* Unlike $K = D$, where affine dependence can create degeneracy, $K = D - 1$ is robust regardless of anchor configuration.
3. *No Added Utility Beyond Isometry:* Recovering up to isometry is sufficient for many applications (e.g., visualization, clustering). Adding more anchors increases identifiability risk without improving downstream performance anymore.

Thus, choosing $K < d$ affinely independent anchors is a principled design choice for privacy-preserving recovery. It: maximizes geometric ambiguity without losing structure, avoids degeneracy from affine dependence, and preserves robustness in both Euclidean and hyperbolic settings. In contrast, using $K = D$ leads to tight constraints, small ambiguity, and easy inference, enabling potential attacks. Our design choice in SENSE and related frameworks deliberately limits anchor access to protect against such risks.

**Note1:** In decentralized or federated settings, adversaries aiming to reconstruct private data using techniques such as model inversion, membership inference, attribute inference, gradient leakage, or reconstruction attacks, gain a significant advantage when the solution space is small and structured (Fredrikson et al., 2015; Shokri et al., 2017; Fredrikson et al., 2014; Zhu et al., 2019; Geiping et al., 2020; Nasr et al., 2019). A constrained set of possible solutions (e.g., a one-dimensional curve or a

mirrored pair of points) reduces uncertainty, making it easier to link gradients or model updates back to the original data. Conversely, a larger and more ambiguous solution space (e.g., a high-dimensional manifold) introduces uncertainty and increases the difficulty of pinpointing a unique inverse mapping. In such cases, adversarial inference becomes more challenging, as multiple plausible candidates exist. Therefore, introducing geometric ambiguity—for instance, by designing the solution space such that $K < d$—can serve as an effective defense mechanism in privacy-sensitive scenarios. Also, non-uniqueness alone is not sufficient: we must consider how large, diverse, and unstructured the solution set is to ensure meaningful privacy.

**Note2:** The *mirror ambiguity problem* arises in localization when distances to a limited set of anchors admit multiple, equally valid solutions that are reflections of each other (Wei et al., 2015; Bose et al., 2017; Hou, 2022; Gerok et al., 2009; Betti et al., 1993; Teunissen, 2017; Saxe, 1979). This occurs especially when the number of anchors $k$ equals the dimensionality $d$. For instance, in 2D with two anchors, the target point lies at the intersection of two circles, yielding two symmetric solutions across the anchor line; in 3D with three anchors, the solution lies on a mirrored circle. In wireless networks, this is referred to as *flip ambiguity*, where measurement noise exacerbates the uncertainty (Wei et al., 2015). Detecting such ambiguity is equivalent to finding a plane intersecting all error spheres of the anchors. The root cause is geometric: distance measurements lack directionality, and $k = d$ anchors do not sufficiently constrain the solution, leading to mirror symmetry. Practical implications include localization errors and navigation failures in robotics and sensor networks. In graph-theoretic terms (Saxe, 1979), the problem corresponds to embedding a weighted graph (nodes as points, edges as distances) into $k$-space. When anchors equal the embedding dimension, the embedding is ambiguous up to reflections, producing multiple valid placements across a plane or hyperplane defined by the anchors.

### A.11 METRIC USED.

- *Cosine Similarity (*CosSim*):* Measures angular similarity between the original NA feature matrix $X'_{\text{NA}} \in \mathbb{R}^{N \times d_h}$ and the reconstructed version $X_{\text{NA}} \in \mathbb{R}^{N \times d_h}$ from SENSE-anchored MDS. Cosine similarity is computed as:

$$\text{CosSim}(X'_{\text{NA}}, X_{\text{NA}}) = \frac{1}{N} \sum_{i=1}^{N} \frac{\langle (X'_{\text{NA}})^{(i)}, X_{\text{NA}}^{(i)} \rangle}{\|(X'_{\text{NA}})^{(i)}\| \cdot \|X_{\text{NA}}^{(i)}\|}$$

  High values (close to 1) indicate strong alignment between original and reconstructed embeddings.
- *Distance Error (DE):* and *F-score (FS):* defined in Section 4.1.
- *Pearson Correlation (*$\rho$*):* Quantifies linear correlation between the original and reconstructed NA–NA distance matrices:

$$\rho = \text{Pearson}(G_{ij}, \widehat{G}_{ij}), \quad \forall i < j$$

  where $G$ and $\widehat{G}$ denote the ground-truth and reconstructed distance matrices respectively. Values close to 1 indicate that the relative distance structure is preserved.
- *Frobenius Norm Error (*$X_{frob}$*):* Measures reconstruction error in the embedding space:

$$X_{\text{frob}} = \frac{\|X_{\text{NA}} - X'_{\text{NA}}\|_F}{\|X'_{\text{NA}}\|_F}$$

  A value of 0 implies perfect reconstruction; higher values suggest increasing deviation.

## A.12    DATASET STATISTICS.

Table 7: Dataset statistics and learning setups grouped by embedding geometry. For hyperbolic, the stats are for *Pointwise* setting.

| Space | Dataset | #Classes | #Datapoints | #Clients (M) | Dimension |
|-------|---------|----------|-------------|--------------|-----------|
| Euclidean | MNIST | 10 | 25000 | 10 | 784 |
| | Fashion-MNIST | 10 | 25000 | 10 | 784 |
| | CIFAR-10 | 10 | 25000 | 5/10 | 1024 |
| | DermaMNIST | 7 | 10015 | 10 | 784 |
| | PneumoniaMNIST | 2 | 5856 | 10 | 784 |
| | RetinaMNIST | 5 | 1600 | 10 | 784 |
| | BreastMNIST | 2 | 780 | 10 | 784 |
| | BloodMNIST | 8 | 17092 | 10 | 784 |
| | OrganCMNIST | 11 | 23583 | 10 | 784 |
| | OrganSMNIST | 11 | 25211 | 10 | 784 |
| | German-Credit | 2 | 1000 | 10 | 20 |
| Hyperbolic | Airport | 4 | 3185 | 3185 | 11 |
| | Amazon | - | 5000 | 5000 | 128 |
| | DBLP | - | 5000 | 5000 | 128 |

## A.13    SYSTEM SPECIFICATIONS

All experiments are conducted on a server equipped with two **NVIDIA RTX A6000** GPUs (48 GB memory each) and an **Intel Xeon Platinum 8360Y** CPU with **1 TB RAM**.

## A.14    VISUALIZATION RESULTS

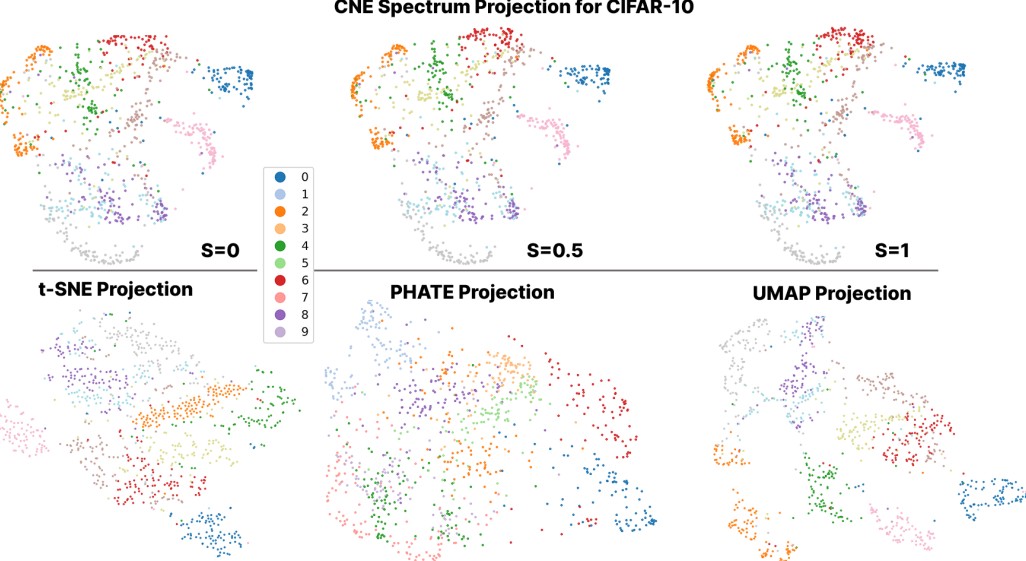

Figure 4: Pointwise setting: CIFAR-10 (1000 non-anchor points, 783 anchors)

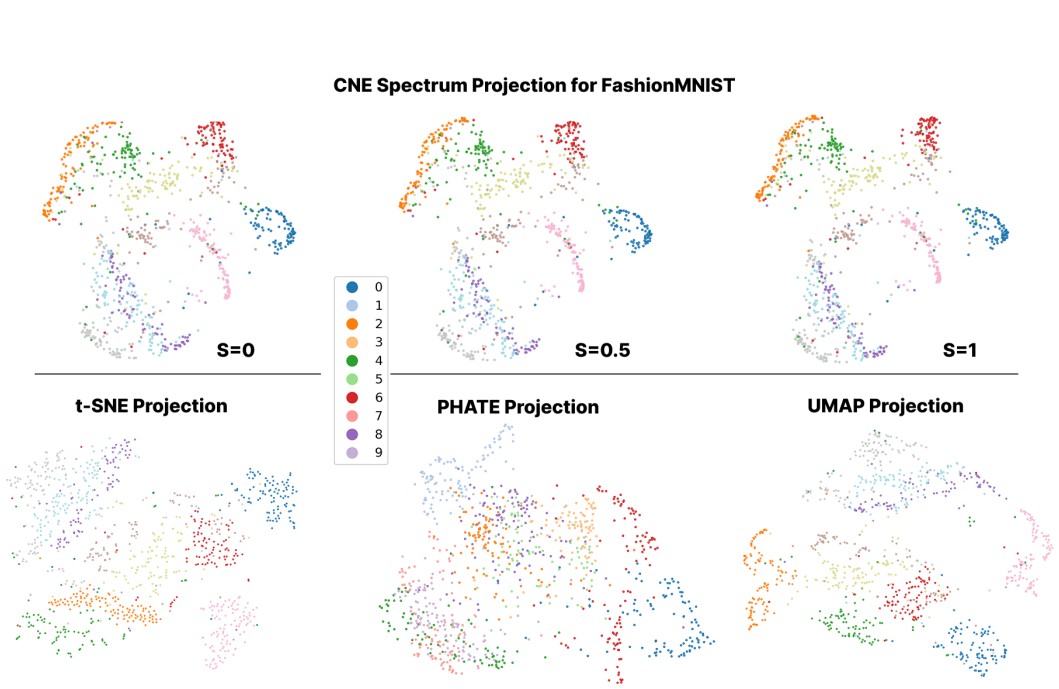

Figure 5: Pointwise setting: FashionMNIST (1000 non-anchor points, 783 anchors)

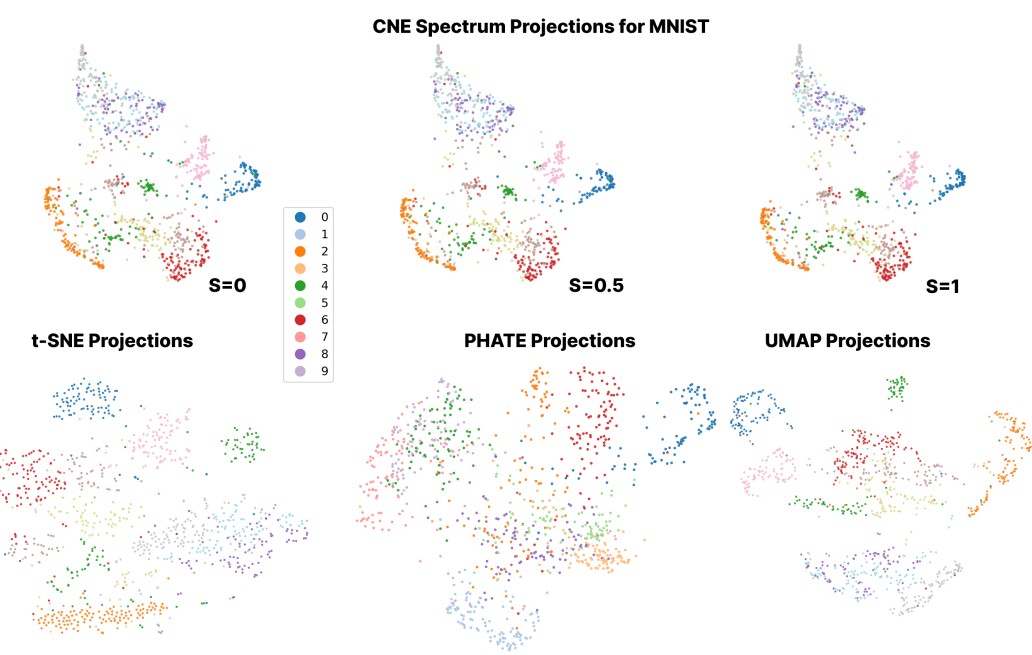

Figure 6: Pointwise setting: MNIST (1000 non-anchor points, 783 anchors)

## A.15 RESULTS.

Table 8: FS and DE across IID, and non-IID balanced and unbalanced splits.

| Data | IID | | Bal | | Unbal | |
|------|-----|-----|-----|-----|-----|-----|
| | FS | DE | FS | DE | FS | DE |
| PNEU. | 0.92 | 0.0052 | 0.87 | 0.0066 | 0.91 | 0.0055 |
| BLOOD | 0.90 | 0.0052 | 0.89 | 0.0051 | 0.90 | 0.0052 |
| BREAST | 0.95 | 0.0092 | 0.92 | 0.0113 | 0.91 | 0.0124 |
| DERMA | 0.96 | 0.0029 | 0.93 | 0.0031 | 0.96 | 0.0029 |
| RETINA | 0.96 | 0.0221 | 0.94 | 0.0272 | 0.96 | 0.0214 |
| ORGANC | 0.80 | 0.0092 | 0.79 | 0.0089 | 0.79 | 0.0092 |
| ORGANS | 0.81 | 0.0089 | 0.80 | 0.0085 | 0.81 | 0.0093 |
| GERMAN | 0.75 | 0.0565 | 0.73 | 0.0621 | 0.72 | 0.0629 |

Table 9: FS and DE under POINTWISE, IID, and NON-IID settings, comparing MULTISITE-FULL and MULTISITE-PARTIAL.

| Dataset | Pointwise | | IID-Full | | IID-Partial | | Non-IID-Full | | Non-IID-Partial | |
|---------|-----------|-----|----------|-----|-------------|-----|--------------|-----|-----------------|-----|
| | FS | DE | FS | DE | FS | DE | FS | DE | FS | DE |
| MNIST | 0.9557 | 0.0057 | 0.8034 | 0.0097 | 0.9266 | 0.0438 | 0.7864 | 0.0101 | 0.9275 | 0.0434 |
| FashionMNIST | 0.9560 | 0.0058 | 0.7586 | 0.0070 | 0.8726 | 0.0153 | 0.7534 | 0.0070 | 0.8754 | 0.0156 |
| CIFAR-10 | 0.9562 | 0.0057 | 0.9303 | 0.0049 | 0.9277 | 0.0044 | 0.9308 | 0.0049 | 0.9380 | 0.0044 |

Table 10: Multisite setting comparison Non-iid unbalanced: Full vs Partial: Evaluation of different methods (Vanilla and SENSE variants) across different metrics.

| Data | Metric | t-SNE | | UMAP | | PHATE | | CNE(s=0) | | CNE(s=0.5) | | CNE(s=1) | |
|------|--------|-------|-------|------|-------|-------|-------|----------|-------|------------|-------|----------|-------|
| | | VAN. | SENSE | VAN. | SENSE | VAN. | SENSE | VAN. | SENSE | VAN. | SENSE | VAN. | SENSE |
| | | — Multisite-Partial Setting — | | | | | | | | | | | |
| | Trust. | 0.9259 | 0.9274 | 0.7447 | 0.7476 | 0.8175 | 0.8174 | 0.8334 | 0.8336 | 0.8322 | 0.8321 | 0.8232 | 0.8244 |
| CIFAR-10 | Cont. | 0.9107 | 0.9391 | 0.8756 | 0.8804 | 0.9369 | 0.9381 | 0.9554 | 0.9552 | 0.9552 | 0.9549 | 0.9565 | 0.9561 |
| | Stead. | 0.8099 | 0.8165 | 0.6904 | 0.6938 | 0.7363 | 0.7349 | 0.7609 | 0.7654 | 0.7619 | 0.7580 | 0.7415 | 0.7487 |
| | Cohes. | 0.4707 | 0.4806 | 0.3725 | 0.3752 | 0.4927 | 0.4857 | 0.4708 | 0.4630 | 0.4716 | 0.4778 | 0.4766 | 0.4793 |
| | | — Multisite-Full Setting — | | | | | | | | | | | |
| | Trust. | 0.9259 | 0.9270 | 0.7447 | 0.7482 | 0.8175 | 0.8168 | 0.8334 | 0.8336 | 0.8322 | 0.8329 | 0.8232 | 0.8247 |
| CIFAR-10 | Cont. | 0.9107 | 0.9364 | 0.8756 | 0.8808 | 0.9369 | 0.9366 | 0.9554 | 0.9553 | 0.9552 | 0.9550 | 0.9565 | 0.9561 |
| | Stead. | 0.8099 | 0.8229 | 0.6904 | 0.6875 | 0.7363 | 0.7357 | 0.7609 | 0.7624 | 0.7619 | 0.7580 | 0.7415 | 0.7464 |
| | Cohes. | 0.4707 | 0.4673 | 0.3725 | 0.3674 | 0.4927 | 0.4831 | 0.4708 | 0.4662 | 0.4716 | 0.4690 | 0.4766 | 0.4811 |
| | | — Pointwise-Full Setting — | | | | | | | | | | | |
| | Trust. | 0.9683 | 0.9659 | 0.9435 | 0.9419 | 0.8488 | 0.8531 | 0.9112 | 0.9123 | 0.9082 | 0.9079 | 0.9021 | 0.9035 |
| | Cont. | 0.9465 | 0.9448 | 0.9379 | 0.9333 | 0.9533 | 0.9527 | 0.9446 | 0.9442 | 0.9458 | 0.9437 | 0.9445 | 0.9442 |
| CIFAR-10 | Stead. | 0.8061 | 0.8081 | 0.7793 | 0.7825 | 0.7111 | 0.7165 | 0.7992 | 0.7878 | 0.7887 | 0.8005 | 0.7808 | 0.7920 |
| | Cohes. | 0.7482 | 0.7672 | 0.7415 | 0.7336 | 0.7431 | 0.7365 | 0.7485 | 0.7451 | 0.7513 | 0.7473 | 0.7435 | 0.7350 |

## A.16 DISCUSSION.

**SENSE in Evolving Distributed Environments.** In dynamic settings, new data points arrive continuously e.g., a hospital admitting a patient, a bank processing a transaction, or a platform onboarding a user. Recomputing the full embedding for each arrival is inefficient and may disrupt global structure. Existing decentralized NE methods (Li et al., 2024; Qiao et al., 2024; Saha et al., 2017; Saha et al.) assume static datasets and lack support for incremental updates, making them unsuitable for streaming environments. SENSE, by contrast, is modular and compatible with out-of-sample embedding methods (Herath et al., 2021; Bengio et al., 2003; Oster et al., 2021). Once the global embedding is constructed via anchor-based completion and NE optimization, it defines a geometry-aware coordinate space that supports new points without full recomputation. Let $\mathbf{X}_{NA} = [\mathbf{x}_1, \ldots, \mathbf{x}_N] \in \mathbb{R}^{N \times d_h}$ be the reconstructed NA embeddings. When a new point $\mathbf{y}$ arrives, we select $K$ existing points as pseudo-anchors $\mathcal{A} = \{a_1, \ldots, a_K\} \subset \mathbf{X}_{NA}$, with coordinates $\mathbf{X}_A = [\mathbf{p}_1, \ldots, \mathbf{p}_K]^\top \in \mathbb{R}^{K \times d_h}$. Given dissimilarities $\{\delta_{l_i y}\}_{i=1}^K$ to these anchors, we compute the embedding $\hat{\mathbf{y}}$ by solving:

$$\hat{\sigma}(\hat{\mathbf{y}}) = \sum_{i=1}^{K} \left( \|\mathbf{p}_i - \hat{\mathbf{y}}\|_2 - \delta_{l_i y} \right)^2. \tag{23}$$

Table 11: IID setting: Evaluation of different dimensionality reduction methods (Vanilla and SENSE variants) across various metrics.

| Data | Metric | t-SNE | | UMAP | | PHATE | | CNE(s=0) | | CNE(s=0.5) | | CNE(s=1) | |
|---|---|---|---|---|---|---|---|---|---|---|---|---|---|
| | | VAN. | SENSE | VAN. | SENSE | VAN. | SENSE | VAN. | SENSE | VAN. | SENSE | VAN. | SENSE |
| PneumoniaMNIST | Trust. | 0.9718 | 0.9700 | 0.7687 | 0.7700 | 0.8573 | 0.8590 | 0.9016 | 0.9026 | 0.8973 | 0.8967 | 0.8837 | 0.8795 |
| | Cont. | 0.9395 | 0.9442 | 0.9145 | 0.9143 | 0.9616 | 0.9598 | 0.9592 | 0.9587 | 0.9591 | 0.9582 | 0.9606 | 0.9598 |
| | Stead. | 0.7840 | 0.7844 | 0.6203 | 0.6272 | 0.7158 | 0.7228 | 0.7554 | 0.7516 | 0.7439 | 0.7424 | 0.7369 | 0.7263 |
| | Cohes. | 0.7031 | 0.6963 | 0.6081 | 0.6272 | 0.6902 | 0.6898 | 0.7013 | 0.7112 | 0.6981 | 0.6970 | 0.7006 | 0.7050 |
| BloodMNIST | Trust. | 0.9628 | 0.9611 | 0.8643 | 0.8633 | 0.8515 | 0.8527 | 0.8847 | 0.8820 | 0.8793 | 0.8820 | 0.8729 | 0.8736 |
| | Cont. | 0.9312 | 0.9280 | 0.9416 | 0.9391 | 0.9444 | 0.9440 | 0.9555 | 0.9558 | 0.9556 | 0.9558 | 0.9553 | 0.9556 |
| | Stead. | 0.7515 | 0.7436 | 0.6899 | 0.6764 | 0.6967 | 0.6871 | 0.7259 | 0.7211 | 0.7228 | 0.7211 | 0.7164 | 0.7133 |
| | Cohes. | 0.7085 | 0.7106 | 0.7233 | 0.7261 | 0.7416 | 0.7469 | 0.7435 | 0.7339 | 0.7329 | 0.7339 | 0.7453 | 0.7462 |
| BreastMNIST | Trust. | 0.9382 | 0.9370 | 0.7599 | 0.7589 | 0.8835 | 0.8774 | 0.8938 | 0.8924 | 0.8939 | 0.8920 | 0.8934 | 0.8924 |
| | Cont. | 0.9452 | 0.9412 | 0.8147 | 0.8174 | 0.9533 | 0.9526 | 0.9450 | 0.9446 | 0.9450 | 0.9445 | 0.9450 | 0.9444 |
| | Stead. | 0.8522 | 0.8514 | 0.5800 | 0.5697 | 0.8056 | 0.8099 | 0.8400 | 0.8400 | 0.8287 | 0.8308 | 0.8317 | 0.8353 |
| | Cohes. | 0.6028 | 0.5987 | 0.4226 | 0.4226 | 0.5639 | 0.5611 | 0.5566 | 0.5605 | 0.5637 | 0.5670 | 0.5532 | 0.5606 |
| DermaMNIST | Trust. | 0.9758 | 0.9762 | 0.7513 | 0.7480 | 0.8726 | 0.8726 | 0.9129 | 0.9118 | 0.9125 | 0.9126 | 0.9017 | 0.9023 |
| | Cont. | 0.9592 | 0.9583 | 0.9134 | 0.9129 | 0.9736 | 0.9729 | 0.9709 | 0.9712 | 0.9707 | 0.9706 | 0.9716 | 0.9714 |
| | Stead. | 0.7995 | 0.7976 | 0.5930 | 0.5945 | 0.7332 | 0.7291 | 0.7726 | 0.7739 | 0.7694 | 0.7638 | 0.7580 | 0.7577 |
| | Cohes. | 0.7294 | 0.7107 | 0.5590 | 0.5618 | 0.7001 | 0.7184 | 0.7339 | 0.7334 | 0.7390 | 0.7373 | 0.7308 | 0.7297 |
| RetinaMNIST | Trust. | 0.9797 | 0.9758 | 0.8777 | 0.8643 | 0.9144 | 0.9038 | 0.9480 | 0.9335 | 0.9469 | 0.9331 | 0.9450 | 0.9313 |
| | Cont. | 0.9669 | 0.9567 | 0.9280 | 0.9232 | 0.9738 | 0.9730 | 0.9718 | 0.9711 | 0.9704 | 0.9700 | 0.9678 | 0.9678 |
| | Stead. | 0.8483 | 0.8479 | 0.6120 | 0.5941 | 0.7618 | 0.7434 | 0.8183 | 0.8140 | 0.8117 | 0.8050 | 0.8105 | 0.8086 |
| | Cohes. | 0.7051 | 0.6963 | 0.5835 | 0.5515 | 0.6980 | 0.6995 | 0.7123 | 0.7074 | 0.7046 | 0.7112 | 0.6831 | 0.7135 |
| OrganCMNIST | Trust. | 0.9608 | 0.9482 | 0.8879 | 0.8815 | 0.8845 | 0.8858 | 0.9149 | 0.9028 | 0.9160 | 0.9039 | 0.9024 | 0.8890 |
| | Cont. | 0.9238 | 0.9413 | 0.9231 | 0.9242 | 0.9696 | 0.9682 | 0.9731 | 0.9683 | 0.9730 | 0.9679 | 0.9738 | 0.9688 |
| | Stead. | 0.6948 | 0.8027 | 0.7575 | 0.7678 | 0.7994 | 0.8058 | 0.8690 | 0.8677 | 0.8788 | 0.8673 | 0.8624 | 0.8593 |
| | Cohes. | 0.4762 | 0.4849 | 0.3335 | 0.3145 | 0.5695 | 0.5153 | 0.4751 | 0.4760 | 0.5268 | 0.5001 | 0.5545 | 0.5166 |
| OrganSMNIST | Trust. | 0.9565 | 0.9421 | 0.8707 | 0.8588 | 0.8766 | 0.8890 | 0.9130 | 0.9026 | 0.9128 | 0.9034 | 0.8991 | 0.8911 |
| | Cont. | 0.9219 | 0.9366 | 0.9248 | 0.9211 | 0.9679 | 0.9717 | 0.9741 | 0.9684 | 0.9732 | 0.9672 | 0.9737 | 0.9679 |
| | Stead. | 0.6793 | 0.7753 | 0.7305 | 0.7513 | 0.7786 | 0.7965 | 0.8609 | 0.8691 | 0.8649 | 0.8745 | 0.8517 | 0.8601 |
| | Cohes. | 0.4856 | 0.4702 | 0.3327 | 0.3316 | 0.5575 | 0.5094 | 0.4838 | 0.4525 | 0.5312 | 0.4889 | 0.5564 | 0.4783 |
| german-credit | Trust. | 0.9771 | 0.9553 | 0.9505 | 0.9330 | 0.8559 | 0.8551 | 0.9380 | 0.9224 | 0.9359 | 0.9140 | 0.9325 | 0.9192 |
| | Cont. | 0.9590 | 0.9434 | 0.9587 | 0.9449 | 0.9482 | 0.9294 | 0.9573 | 0.9448 | 0.9573 | 0.9429 | 0.9564 | 0.9432 |
| | Stead. | 0.8603 | 0.8251 | 0.8342 | 0.7907 | 0.7500 | 0.7228 | 0.8414 | 0.7954 | 0.8416 | 0.7883 | 0.8401 | 0.7944 |
| | Cohes. | 0.6810 | 0.6895 | 0.6542 | 0.6413 | 0.6712 | 0.6640 | 0.6465 | 0.6651 | 0.6577 | 0.6675 | 0.6624 | 0.6550 |

Table 12: Non-IID (balanced) setting: Evaluation of different methods (Vanilla and SENSE variants) across different metrics.

| Data | Metric | t-SNE | | UMAP | | PHATE | | CNE(s=0) | | CNE(s=0.5) | | CNE(s=1) | |
|---|---|---|---|---|---|---|---|---|---|---|---|---|---|
| | | VAN. | SENSE | VAN. | SENSE | VAN. | SENSE | VAN. | SENSE | VAN. | SENSE | VAN. | SENSE |
| PneumoniaMNIST | Trust. | 0.9566 | 0.9483 | 0.8806 | 0.8658 | 0.8909 | 0.8937 | 0.9430 | 0.9393 | 0.9372 | 0.9343 | 0.9226 | 0.9168 |
| | Cont. | 0.9228 | 0.9278 | 0.9031 | 0.9114 | 0.9776 | 0.9732 | 0.9683 | 0.9678 | 0.9690 | 0.9686 | 0.9704 | 0.9695 |
| | Stead. | 0.6952 | 0.7165 | 0.6007 | 0.6211 | 0.7146 | 0.7244 | 0.7778 | 0.7737 | 0.7694 | 0.7692 | 0.7622 | 0.7579 |
| | Cohes. | 0.6377 | 0.6815 | 0.6205 | 0.6070 | 0.6650 | 0.6771 | 0.7259 | 0.7162 | 0.7240 | 0.7145 | 0.7172 | 0.7336 |
| BloodMNIST | Trust. | 0.9304 | 0.9292 | 0.8902 | 0.8796 | 0.8640 | 0.8633 | 0.9003 | 0.8972 | 0.8959 | 0.8944 | 0.8862 | 0.8856 |
| | Cont. | 0.9020 | 0.9029 | 0.9385 | 0.9390 | 0.9510 | 0.9492 | 0.9618 | 0.9611 | 0.9620 | 0.9614 | 0.9622 | 0.9614 |
| | Stead. | 0.7060 | 0.7017 | 0.6815 | 0.6927 | 0.6812 | 0.6927 | 0.7531 | 0.7505 | 0.7466 | 0.7442 | 0.7536 | 0.7395 |
| | Cohes. | 0.6781 | 0.6761 | 0.7210 | 0.7096 | 0.7620 | 0.7540 | 0.7441 | 0.7603 | 0.7472 | 0.7335 | 0.7561 | 0.7603 |
| BreastMNIST | Trust. | 0.9643 | 0.9657 | 0.8476 | 0.8562 | 0.9188 | 0.9241 | 0.9403 | 0.9422 | 0.9385 | 0.9418 | 0.9383 | 0.9415 |
| | Cont. | 0.9632 | 0.9658 | 0.8567 | 0.8408 | 0.9587 | 0.9671 | 0.9604 | 0.9594 | 0.9598 | 0.9590 | 0.9599 | 0.9591 |
| | Stead. | 0.8331 | 0.8370 | 0.5159 | 0.5081 | 0.7585 | 0.7913 | 0.8712 | 0.8742 | 0.8684 | 0.8616 | 0.8691 | 0.8675 |
| | Cohes. | 0.6174 | 0.6018 | 0.3677 | 0.3741 | 0.5187 | 0.5165 | 0.5254 | 0.5667 | 0.5265 | 0.5413 | 0.5200 | 0.5485 |
| DermaMNIST | Trust. | 0.9545 | 0.9467 | 0.8253 | 0.8048 | 0.8963 | 0.8961 | 0.9335 | 0.9351 | 0.9292 | 0.9327 | 0.9147 | 0.9167 |
| | Cont. | 0.9403 | 0.9284 | 0.8977 | 0.8895 | 0.9825 | 0.9815 | 0.9742 | 0.9734 | 0.9743 | 0.9733 | 0.9761 | 0.9756 |
| | Stead. | 0.7304 | 0.7148 | 0.5608 | 0.5428 | 0.7327 | 0.7295 | 0.7901 | 0.7909 | 0.7834 | 0.7841 | 0.7751 | 0.7743 |
| | Cohes. | 0.6493 | 0.6484 | 0.5159 | 0.5152 | 0.6867 | 0.6726 | 0.6993 | 0.6976 | 0.6976 | 0.7128 | 0.6902 | 0.7012 |
| RetinaMNIST | Trust. | 0.9749 | 0.9743 | 0.8933 | 0.8829 | 0.9228 | 0.9227 | 0.9522 | 0.9523 | 0.9492 | 0.9519 | 0.9497 | 0.9495 |
| | Cont. | 0.9627 | 0.9616 | 0.9289 | 0.9152 | 0.9752 | 0.9729 | 0.9720 | 0.9713 | 0.9712 | 0.9700 | 0.9670 | 0.9675 |
| | Stead. | 0.8447 | 0.8380 | 0.6155 | 0.6174 | 0.7534 | 0.7559 | 0.8224 | 0.8172 | 0.8134 | 0.8189 | 0.8123 | 0.8046 |
| | Cohes. | 0.7140 | 0.7283 | 0.5785 | 0.5648 | 0.7189 | 0.6836 | 0.7292 | 0.7005 | 0.7092 | 0.6938 | 0.7039 | 0.6849 |
| OrganCMNIST | Trust. | 0.9489 | 0.9271 | 0.8975 | 0.8888 | 0.9005 | 0.8984 | 0.9235 | 0.9132 | 0.9232 | 0.9126 | 0.9140 | 0.8994 |
| | Cont. | 0.9210 | 0.9082 | 0.9232 | 0.9185 | 0.9737 | 0.9719 | 0.9756 | 0.9715 | 0.9750 | 0.9710 | 0.9760 | 0.9717 |
| | Stead. | 0.6365 | 0.7142 | 0.7462 | 0.7290 | 0.8038 | 0.7909 | 0.8611 | 0.8724 | 0.8660 | 0.8745 | 0.8621 | 0.8640 |
| | Cohes. | 0.4862 | 0.4913 | 0.3249 | 0.3191 | 0.5088 | 0.5154 | 0.5338 | 0.4980 | 0.5266 | 0.4974 | 0.4908 | 0.5282 |
| OrganSMNIST | Trust. | 0.9383 | 0.9093 | 0.8954 | 0.8861 | 0.9054 | 0.9071 | 0.9269 | 0.9190 | 0.9291 | 0.9194 | 0.9172 | 0.9092 |
| | Cont. | 0.9164 | 0.8881 | 0.9168 | 0.9255 | 0.9774 | 0.9758 | 0.9796 | 0.9746 | 0.9786 | 0.9741 | 0.9788 | 0.9741 |
| | Stead. | 0.5896 | 0.6154 | 0.6315 | 0.6953 | 0.7784 | 0.7963 | 0.8591 | 0.8684 | 0.8560 | 0.8634 | 0.8411 | 0.8523 |
| | Cohes. | 0.5109 | 0.5108 | 0.3441 | 0.3665 | 0.5642 | 0.5278 | 0.5079 | 0.4878 | 0.5461 | 0.5021 | 0.5487 | 0.5001 |
| german-credit | Trust. | 0.9752 | 0.9575 | 0.9511 | 0.9301 | 0.8552 | 0.8508 | 0.9403 | 0.9211 | 0.9380 | 0.9172 | 0.9350 | 0.9176 |
| | Cont. | 0.9581 | 0.9418 | 0.9606 | 0.9427 | 0.9481 | 0.9240 | 0.9576 | 0.9470 | 0.9575 | 0.9463 | 0.9571 | 0.9460 |
| | Stead. | 0.8567 | 0.8267 | 0.8350 | 0.7850 | 0.7398 | 0.7023 | 0.8484 | 0.8063 | 0.8475 | 0.8016 | 0.8405 | 0.8020 |
| | Cohes. | 0.6795 | 0.6837 | 0.6488 | 0.6509 | 0.6870 | 0.6828 | 0.6620 | 0.6834 | 0.6557 | 0.6676 | 0.6564 | 0.6653 |

Here, $\delta_{l_i y}$ is the dissimilarity in the original space, and $\|\mathbf{p}_i - \hat{\mathbf{y}}\|_2$ is the distance in the embedding space. Only $\hat{\mathbf{y}}$ is optimized, anchors remain fixed. Since $K < d_h$, exact recovery is impossible (Theorems 3.1, 3.2), ensuring privacy. This lightweight optimization requires no raw data and supports real-time integration, making SENSE well-suited for scalable, privacy-constrained systems.

**Scalability and Computational Complexity.** In addition to the 14 standard DR datasets (Sec. 4.1), we evaluate SENSE on two large-scale benchmarks: Tiny ImageNet (Le & Yang, 2015) ($\sim$90k NAs, with 512D features extracted using an ImageNet-pretrained ResNet-34 (He et al., 2016)) and Street View House Numbers (Netzer et al., 2011) (SVHN, $\sim$80k NAs, $d_h = 512$). These experiments are conducted for *Multisite setting*, where we distributed the NA samples to 10 clients in non-IID unbalanced scenarios. Results in Table 13 demonstrate that SENSE maintains strong performance even at this scale, with runtimes of only $\sim$12–14 seconds per iteration.

Table 13: SENSE performance compared to DR baselines on Tiny ImageNet and SVHN. Runtime is averaged per iteration.

| Data | Metric | t-SNE VAN | SENSE | UMAP VAN | SENSE | PHATE VAN | SENSE | CNE(0) VAN | SENSE | CNE(0.5) VAN | SENSE | CNE(1) VAN | SENSE | Runtime |
|---|---|---|---|---|---|---|---|---|---|---|---|---|---|---|
| TinyImageNet | Trust | 0.9245 | 0.9341 | 0.7748 | 0.7330 | 0.7480 | 0.7392 | 0.7682 | 0.7402 | 0.7960 | 0.7637 | 0.7717 | 0.7467 | 13.72s |
| | Cont | 0.9107 | 0.9064 | 0.9334 | 0.9140 | 0.9359 | 0.9029 | 0.9396 | 0.9268 | 0.9350 | 0.9244 | 0.9411 | 0.9294 | |
| | Stead | 0.8099 | 0.8229 | 0.5703 | 0.5755 | 0.5550 | 0.6248 | 0.5848 | 0.6205 | 0.5986 | 0.6298 | 0.5807 | 0.6135 | |
| | Cohes | 0.7680 | 0.7548 | 0.8305 | 0.6685 | 0.8340 | 0.7439 | 0.8344 | 0.7223 | 0.8296 | 0.7160 | 0.8403 | 0.7322 | |
| SVHN | Trust | 0.9822 | 0.9801 | 0.8973 | 0.8914 | 0.8825 | 0.8819 | 0.8910 | 0.8900 | 0.9033 | 0.9068 | 0.8939 | 0.8964 | 13.50s |
| | Cont | 0.9619 | 0.9630 | 0.9749 | 0.9701 | 0.9759 | 0.9665 | 0.9778 | 0.9745 | 0.9787 | 0.9648 | 0.9787 | 0.9646 | |
| | Stead | 0.7234 | 0.7200 | 0.6545 | 0.6664 | 0.6426 | 0.6739 | 0.6540 | 0.6968 | 0.6705 | 0.7027 | 0.6556 | 0.7134 | |
| | Cohes | 0.8362 | 0.8349 | 0.8430 | 0.7160 | 0.8544 | 0.7834 | 0.8503 | 0.7935 | 0.8493 | 0.6960 | 0.8576 | 0.7535 | |

We also provide detailed runtime and complexity analysis. With $N$ NA points and $K$ anchors, Anchored-MDS in Sec 2 has complexity $\mathcal{O}(K^2 d_h + KN d_h)$, efficient as $K \ll N$ and $K < d_h$ (for privacy). We use the fastest variants of different global low-dimensional embedding methods in our framework, as this is the second stage in the pipeline. Notation: here k denotes the number of neighbors considered per point (in the attractive force calculation), $d_h$ is the embedding dimension, $n$ are the number of data points (samples) and $m$ are the number of negative (repulsive) samples per positive interaction. 1) Fast NE methods: Van-t-SNE takes $\mathcal{O}(n^2 d_h)$ (van der Maaten & Hinton, 2008), Barnes-Hut t-SNE (BH-t-SNE) and FIt-SNE (FFT-based interpolation) takes $\mathcal{O}(kn \log n \cdot 2^{d_h})$ and $\mathcal{O}(kn \cdot 2^{d_h})$ respectively. 2) The contrastive neighbor embedding: These methods only sample $nm$ repulsive interactions per epoch (instead of all pairs) with complaexity $\mathcal{O}(kmn d_h)$, which scales linearly with embedding dimension $d_h$. This also includes NC-t-SNE/UMAP with contrastive loss runs in $\mathcal{O}(km d_h)$ with $m \ll n$ repulsive samples per epoch (Damrich et al., 2023).

We also provide empirical results on the different stages of the pipeline. Table 14 reports empirical results for individual stages of the SENSE pipeline.

Table 14: Runtime for SENSE pipeline stages: stage1 = incomplete matrix, stage2 = matrix completion.

| Data | NA | K | Stage1(s) | Stage2(s) | Total(s) | FS | DE |
|---|---|---|---|---|---|---|---|
| BloodMNIST | 1k | 100 | 0.33 | 2.22 | 2.55 | 0.86 | 0.02 |
| | | 1k | 0.34 | 8.11 | 8.45 | 0.94 | 0.01 |
| | 5k | 100 | 0.46 | 39.33 | 39.79 | 0.81 | 0.00 |
| | 2k | 50 | 0.59 | 54.83 | 55.42 | 0.92 | 0.01 |

**Curated Privacy Example.** To illustrate the privacy guarantees of SENSE, we constructed a curated dataset containing sensitive attributes (e.g., age) and categorical features (e.g., gender, occupation). The setup involves five clients ($C_1$–$C_5$), each with five attributes ($d = 5$), and four reference anchors ($K = 4 < d$). Embeddings are computed exclusively from client–anchor distances, without direct access to raw features. As shown in Table 15, the resulting embeddings preserve structural relationships while preventing recovery of private attributes, thereby empirically validating the privacy-preserving nature of SENSE.

**Why SENSE Avoids Noise-Based Privacy.** Noise injection is a common privacy mechanism, but SENSE is built on a different principle. The goal is not to obscure the distance map, but to

Table 15: Original private attributes vs. SENSE embeddings. Sensitive details are not exposed by the embeddings.

| Original Data | $x_1$ | $x_2$ | $x_3$ | $x_4$ | $x_5$ |
|---|---|---|---|---|---|
| $C_1$ | 1500 | 25 | 3 | 1 | 10 |
| $C_2$ | 2000 | 35 | 2 | 0 | 15 |
| $C_3$ | 1745 | 28 | 2 | 0 | 18 |
| $C_4$ | 1620 | 32 | 1 | 1 | 13 |
| $C_5$ | 1200 | 45 | 3 | 1 | 12 |
| **SENSE Emb** | $x_1$ | $x_2$ | $x_3$ | $x_4$ | $x_5$ |
| $C_1$ | 80.630 | 26.896 | 13.795 | 96.939 | -39.321 |
| $C_2$ | -189.837 | -28.557 | -61.648 | -272.603 | 141.184 |
| $C_3$ | -58.064 | 2.598 | -26.164 | -79.338 | 49.985 |
| $C_4$ | -5.492 | -2.856 | 6.935 | 12.915 | -12.891 |
| $C_5$ | 210.016 | 14.741 | 75.140 | 328.458 | -172.165 |

preserve inter-client similarity while ensuring that high-dimensional raw features remain private. In this setting, noise-based privacy is problematic for two reasons. First, robustness to injected noise is inherently unreliable in decentralized DR, where incomplete similarity information amplifies perturbations. Second, our experiments show that even mild noise sharply degrades embedding quality, making noise-based approaches unstable and costly. We evaluated two scenarios: (i) noise added to client–anchor distance vectors, and (ii) Gaussian perturbations applied directly to raw features.

**Empirical Evaluation 1: Noise Added to Anchor-NA Distance Vectors.** We injected noise into the $F$ block of Eq. 8, corresponding to client–anchor distances, under the *Pointwise* setting (each client has a single NA). A random fraction of clients was selected, and noise was added to their anchor distance vectors. Results on *Iris* and *Seeds* (Table 16) show a clear trade-off: as the fraction of noisy clients increases, F-score (FS) drops sharply while Distance Error (DE) rises, confirming the fragility of noise-based privacy in SENSE.

Table 16: Effect of noise injection into $F$ matrix: FS decreases and DE increases as noise fraction increases.

| Data | Metric | 0% | 5% | 10% | 20% | 50% |
|---|---|---|---|---|---|---|
| Iris | FS | 0.8659 | 0.8403 | 0.8104 | 0.7000 | 0.6167 |
| | DE | 0.0425 | 0.0735 | 0.0929 | 0.1528 | 0.1828 |
| Seeds | FS | 0.9745 | 0.9390 | 0.9038 | 0.7902 | 0.6589 |
| | DE | 0.0102 | 0.1251 | 0.1760 | 0.2471 | 0.3702 |

**Empirical Evaluation 2: Noise to Raw Features.** We also injected Gaussian noise directly into raw features (e.g., pixel-level perturbations on MNIST) using the *Multisite* setting of SENSE with $K = d_h$, 10 clients, and 100 NAs. In the noise-based variant (NS), element-wise zero-mean Gaussian noise with varying standard deviations (scaled to the data range) was applied. As shown in Table 17, even mild perturbations lead to sharp drops in FS and significant increases in DE, confirming that noise-based privacy comes at a substantial cost to utility compared with SENSE without noise. In real-world domains such as healthcare or finance, where precision is critical, this trade-off is unacceptable.

Table 17: Performance of SENSE versus noise-based variants (NS) with Gaussian perturbation of raw features.

| Method | DE | FS |
|---|---|---|
| SENSE | 0.006023 | 0.974955 |
| NS1 | 0.074474 | 0.946197 |
| NS2 | 0.268156 | 0.887382 |
| NS3 | 0.535183 | 0.801178 |
| NS4 | 0.843552 | 0.691761 |
| NS5 | 1.175783 | 0.573909 |

Across both evaluations, noise-based privacy consistently degraded utility without providing stronger guarantees. These findings highlight why SENSE achieves privacy not through artificial corruption, but through geometric underdetermination: the structure needed for downstream tasks is preserved, while raw features remain unrecoverable. In domains such as healthcare and finance, where precision is critical, this distinction makes noise-based privacy an impractical choice.

