# OpenReview forum: "SENSE: $\underline{\text{SEN}}$sing Similarity $\underline{\text{SE}}$eing Structure"
_ICLR.cc/2026/Conference — Submitted to ICLR 2026_

### Official Review · Reviewer_seDM · 2025-10-31

[review text omitted: it was posted to a different submission]

---

> ### Author Response · Authors · 2025-11-16
> **Response to the Reviewer: Review does not correspond to the paper we submitted.**
>
> Thank you for taking the time to review our submission. However, **we are concerned that your review does not correspond to the paper we submitted.**
>
> 1. The review cites a title “Structured Embedding via Neighborhood Smoothing and Entropy regularization (SENSE)” and describes a method based on neighborhood smoothing, a contrastive loss, and an entropy regularizer. Our paper is titled “SENSE: Sensing Similarity, Seeing Structure” and proposes a privacy-preserving decentralized distance-based embedding framework with anchors and structured matrix completion.
>
> 2. The datasets and benchmarks you mention (e.g., Cora, Citeseer, Pubmed, Ogbn-Arxiv, and baselines like DeepWalk, Node2Vec, GraphSAGE) do not appear anywhere in our experiments, which are instead on image, medical, tabular, and hyperbolic graph datasets.
>
> 3. The additional baselines you request (e.g., GraphCL, DGI, GCA) are GNN contrastive/self-supervised methods that are not part of our problem setting and are not discussed in the paper.
>
> 4. The theoretical assumptions you critique (smoothness and neighborhood homogeneity) are also not made in our manuscript.
>
> ## **Because so many central aspects of the review (title, method description, datasets, baselines, and assumptions) do not match our work, we believe there has been a mistake and this review may have been inadvertently written for a different paper with a similar acronym or generated from a template that was not aligned with our submission.**
>
> **We would be very grateful if you could re-check the manuscript for submission 13266, “SENSE: Sensing Similarity, Seeing Structure,” and, if appropriate, revise your assessment based on the actual content of our paper.**

---

### Official Review · Reviewer_o7FR · 2025-10-31

**Soundness:** 2
**Presentation:** 1
**Contribution:** 1
**Rating:** 0
**Confidence:** 4

**Summary:**

The paper proposes SENSE, a dimensionality reduction framework which is designed for a decentralized, privacy-sensitive setting. The authors wish to avoid sharing raw data by having local parties compute distances to a shared set of "anchor" points. These anchor distances are then used to reconstruct a global similarity matrix using matrix completion. This completed matrix is then used to learn a low-dimensional embedding and the method is applicable to Euclidean and hyperbolic distances.

**Strengths:**

Dimensionality reduction is of broad interest to the ML community, especially in settings where communication costs of privacy are important.

**Weaknesses:**

To me paper has some flaws in its motivation, problem formulation, and technical exposition, which make it hard for me to evalute it's contributions.

- First, the paper starts by saying "most similarity-based DR approaches assume centralized access to all pairwise similarities". This seems very inaccurate and misleading to me as many established DR methods do not require this. For example, widely used methods such as PCA or random projections (e.g. using Johnson Lindenstrauss based mappings) can be very easily computed in a decentralized or distributed fashion. For example, JL maps can be generated from a shared random seed meaning none of the data has to be shared, while still guaranteeing that the resulting projections can be compared in terms of preserving pairwise distances. Another class of mappings that contradicts what seems to be the central thesis of the paper are learned mappings (e.g. autoencoders or neural networks trained on prior public data or public models). They learn an explicit mapping to a lower dimensional space and once trained, the network on public data, the network can be locally accessed by any party.

- The other main weakness of the paper is that it lacks a well defined formal problem setting. The paper claims to handle privacy, but does not have a formal privacy statement (for ex differential privacy is a formal privacy gurantee). The paper claims to work in a decentralized setting, but does not define a formal communication model between parties holding pieces of the data. While these are practical considerations, it is difficult to understand the merits of the proposed method if privacy/communication is not formally defined. For example, how is the data partitioned across parties? What are the exact communication model? Is there a central server where the parties can interact with? What is the adversarial model for privacy? In terms of privacy, the main statement is essentially that points cannot be exactly reconstructed unless some number of distances are revealed, but this is almost a trivial observation from parameter counting. However, even if points cannot be reconstructed exactly, there is always some leakage of information even if a single distance is observed! For ex, even if you know that a 'private' point has a certain distance to a fixed point, it leaks some about of non trivial 'information' about the private point. That is why a formal privacy model is necessary! IT is also impossible to compare the paper's 'privacy' guarantee to any other method in a meaningful way since it is not formally defined.

- The paper also does not give a very clear description of the proposed algorithm. E.g. there is no algorithm block or a clear step by step guide, clearly explaining the process from the input data (at each local party) to the final low-dimensional representation. What are the precise assumptions? What exactly is the computational problem being solved? The method appears to be a vague combination of arbitrary steps (matrix completion, contrastive learning) without a clear unifying objective. In addition, there are no formal guarantees about the proposed method. They claim to be minimizing a reconstruction error (Eq. 5), but no precise bounds are provided for this error. While a reconstruction error is defined (Equation 5), no theoretical bounds are provided for this error. What is its "natural" scale? How does it depend on the number of anchors, data points, or assumptions on the geometry? Does the learned embedding preserve distances, neighborhood structure, or any other geometric property with any guarantee?

- The paper's claim about efficiencuy are also very unclear. Note that projection based methods such as PCA or JL transforms have a linear complexity on the number of points, but the submission is proposing a constratinve learning objective, which typically requires knowledge of pairwise distances, which is an $O(n^2)$ operation. Thus, the paper does not adequately explain how its method (which seems to require a dense $n \times n$ matrix after completion) is more efficient than projection-based methods.


- In terms of novelty, while the proposed idea of the paper could be novel, but in its current form, the contribution is entirely unclear.

Thus overall, I cannot identify any single community that would be excited about this paper. The privacy community would clearly be very uncomfortable with the lack of formal privacy guarantees. The federated learning community would not be satisfied with the lack of a formal communication model. Finally, empirical ML researchers would not be satisfied with the lack of comparison to very sensible baselines e.g. model based or projection based methods.

**Questions:**

See above.

---

> ### Author Response · Authors · 2025-11-17
> **(Part 1/6) Response to the Reviewer: "most similarity-based DR approaches assume centralized access to all pairwise similarities"**
>
> We thank the reviewer for the thoughtful feedback. We agree that parts of the introduction can be phrased more carefully, and we will revise the text accordingly. However, we respectfully disagree with several of the core criticisms. Many of the requested clarifications are already present in Secs. 2–3 and App. A.8–A.10; in the revision, we will make these elements more prominent and explicit. Below, we address the reviewer’s points in turn (we will elaborately clarify each concern).
>
> **A1:** Our intention was to contrast SENSE with pairwise-similarity-based DR methods such as t-SNE, UMAP, PHATE, CNE, and hyperbolic CoSNE, which indeed assume access to a (nearly) complete similarity/distance matrix or affinity graph in a central location. This is exactly what makes them hard to deploy in decentralized settings where cross-client distances are unavailable or non-shareable. We fully agree that PCA and JL-based mappings are important exceptions (e.g., [1-3]). These methods, however, solve a different problem from SENSE:
>
> * PCA/JL preserves global linear structure or Euclidean distances (up to small multiplicative distortion) in a fixed low-dimensional subspace. SENSE instead reconstructs a global similarity geometry from a structuredly incomplete distance matrix and then feeds that reconstructed geometry into any neighbor-embedding/similarity-based DR method (t-SNE, UMAP, PHATE, CNE, hyperbolic CoSNE, etc.)
> * Federated PCA/JL assumes each client has local feature vectors and sends compressed linear statistics (covariance sketches, projected features) to a server. SENSE never shares raw features or embeddings: clients communicate only distances to a small public anchor set and (optionally) a few intra-client distances, and the server works purely in the distance domain.
> * PCA/JL produces a single Euclidean subspace. SENSE reconstructs a global pairwise distance matrix that can subsequently be used with Euclidean or hyperbolic DR or any other distance-based method, enabling, for example, hyperbolic embeddings for hierarchical data, where linear PCA/JL are known to be inadequate.
>
> In principle, one could first apply a local JL or federated PCA step (to obtain lower-dim features) and then run SENSE on top of those projected features, making SENSE complementary to these methods rather than a competitor. *While we already write that “most similarity-based DR approaches assume centralized access to all pairwise similarities,” we will clarify this distinction more explicitly.*
>
> **On learned mappings (autoencoders / public neural networks):** We agree that once an encoder $f:\mathbb{R}^{d_h} \to \mathbb{R}^{d_\ell}$ has been trained on suitable data, each client can apply $f$ locally and share only low-dimensional features. However, this paradigm differs from the setting SENSE is designed for.
>
> * Learned encoders typically require a large, representative public dataset (often with labels) and a sufficiently expressive model family. In many of our target scenarios (multi-site medical or financial data), such a dataset is unavailable or poorly matched to the private domains. SENSE assumes much less: it only requires a small public anchor set and uses distances to these anchors; anchors can be drawn from limited public data or synthetic prototypes (App. A.8), without needing a strong public encoder.
>
> * In encoder-based approaches, clients share feature embeddings. In SENSE, clients never send features or embeddings: they send only distances to anchors. The server assembles these into a block-structured, partially observed distance matrix and performs structured distance matrix completion to reconstruct a global geometry before learning low-dim embeddings. A pretrained encoder does not address this structured distance-completion problem under missing cross-client similarities.
>
> * A public encoder defines a fixed Euclidean feature space and ignores the actual observation mask over distances. SENSE is explicitly built around the observation mask $\Omega$ (which entries of $E, F, G$ are visible under the 4 regimes) and enforces consistency only with those observed distances. This allows SENSE to operate in regimes with strict distance-only communication and heavy structured missingness.
>
> * Finally, SENSE’s reconstruction layer is geometry-agnostic: Standard public encoders are almost always Euclidean and do not provide a principled way to reconstruct or control the pairwise distance geometry in either Euclidean or hyperbolic space. Our geometric privacy guarantees (Theorems 3.1–3.2) are also tied to this distance-only protocol; they no longer apply if clients directly share encoder outputs.
>
> References:
> [1] Grammenos et al., "Federated Principal Component Analysis," 2020.
> [2] Cruz et al., "Distributed Learning for Autonomous Vehicles in Communication-Constrained Environments," IEEE CASE, 2024.
> [3] Deb et al., "Beyond Johnson–Lindenstrauss: Uniform Bounds for Sketched Bilinear Forms," 2025.

---

> ### Author Response · Authors · 2025-11-17
> **(Part 2/6) Response to the Reviewer: "The other main weakness of the paper is that it lacks a well defined formal..."**
>
> **A2:** **(1) Formal decentralized setting and data partition:** For this, please see sec2, 3, Fig1 & Appendix. For elaborated details on anchor generation check Appen.A.8.
>
> **Communication protocol:** Each client $m$ computes i) distances from its NAs to all available anchors & ii) optionally a small set of local NA-NA distances. Clients send only these distances to the server in a *single non-interactive round.* iii) The server aggregates them into the observed entries of $F$ & $G$ according to $\Omega$. Raw feature vectors thereof are never transmitted.
>
> **Adversary model:** The server (& any eavesdropper observing the client-server channel) is honest-but-curious it follows the protocol but may attempt to reconstruct private features or infer sensitive attributes from the observed distances & the resulting embeddings. Clients do not collude. This matches the standard FL model with distance messages instead of gradient updates. We explicitly do not claim protection against colluding clients or active adversaries; this is a limitation we can state clearly.
>
> **(2) Our privacy notion** has 3 components: (i) Deterministic geometric non-identifiability: the released distances must not uniquely determine the high-dim private features. (ii) Information-theoretic leakage bound: we quantify how much information the distances reveal about the private features. (iii) Empirical attack-based evaluation.
>
> **(i) Deterministic non-identifiability (Thm):** For a single NA point $x\in \mathcal{M}$ in $\mathbb{R}^{d_h}$, consider the distance map $\phi(x) = (\\|x - p_1\\|^2,\dots,\\|x - p_K\\|^2)^\top \in \mathbb{R}^K,$ where $P = \\{p_1,\dots,p_K\\}$ are the anchors. This is exactly the information that any fixed NA point reveals in SENSE.
>
> **Theorem (already implicit in Thm.3.1, 3.2 & lemma):** Let $P \subset \mathbb{R}^{d_h}$ be a set of anchors whose coordinate matrix has $\operatorname{rank}(P) = K < d_h.$ Define the distance map $\phi :\mathbb{R}^{d_h} \to \mathbb{R}^K, \quad \phi(x) = \big(\\|x - p_1\\|^2,\dots,\\|x - p_K\\|^2\big)^\top$ & let $D\phi(x) \in \mathbb{R}^{K \times d_h}$ denote its Jacobian matrix at $x$, whose $k$-th row is $\nabla_x \phi_k(x)^\top$. Then for every regular value $y \in \mathbb{R}^K$ of $\phi$ (i.e., such that for all $x \in \phi^{-1}(y)$ we have $\operatorname{rank}(D\phi(x)) = K$), the preimage $\phi^{-1}(y)$ is a smooth submanifold of $\mathbb{R}^{d_h}$ of dim $d_h - K$. In particular, for any $x \in \mathbb{R}^{d_h}$ such that $y = \phi(x)$ is a regular value, there exists an uncountable $(d_h-K)$-dim continuum of distinct points $x' \in \mathbb{R}^{d_h} \setminus \\{x\\}$ with $\phi(x') = \phi(x),$ so exact recovery of $x$ from its anchor distances $\phi(x)$ is impossible. Thus, when $K < d_h$, the anchor distance map $\phi$ is non-injective and the set of points consistent with a given distance vector has (at least) dim $d_h - K$.
>
> Proof: Will provide if asked (due to space constraints not putting here).
>
> **(ii) We now quantify average-case information leakage** in a simplified probabilistic model by locally linearizing distance map around a point. Fix a point $x_0 \in \mathbb{R}^{d_h}$ and let $A= D\phi(x_0) \in \mathbb{R}^{K \times d_h}$ be the Jacobian of the anchor-distance map $\phi$ at $x_0$. We then approximate the distance channel in a neighborhood of $x_0$ by a linear Gaussian model. Let $X \in \mathbb{R}^{d_h}$ be a random private feature vector with $X \sim \mathcal{N}(0,\Sigma_x), \quad \Sigma_x \succeq 0,$ & consider the linearized observation model $Y = A X + N,$
> where $N \sim \mathcal{N}(0,\Sigma_n)$, $\Sigma_n \succ 0$, is independent Gaussian noise. Then $X$ & $Y$ are jointly Gaussian and the mutual information is $I(X;Y)= \tfrac{1}{2}\log\det\bigl(I + \Sigma_x A^\top \Sigma_n^{-1} A\bigr),$
> see [1-2]. In isotropic case $\Sigma_x = \sigma_x^2 I_{d_h}, \quad \Sigma_n = \sigma_n^2 I_K, $ let $\lambda_1,\dots,\lambda_r$ be the nonzero eigenvalues of $A^\top A$, where $r = \mathrm{rank}(A) \le K$. Then
> $$
> I(X;Y)
> = \tfrac{1}{2}\sum_{j=1}^r \log\Bigl(1 + \tfrac{\sigma_x^2}{\sigma_n^2}\lambda_j\Bigr).
> $$
> Moreover, if spectral norm of $A$ is bounded, i.e., $\lambda_j \le \lambda_{\max} \quad \text{for all } j,$ we obtain upper bound
> $$
> I(X;Y)
> \le
> \tfrac{r}{2}\log\Bigl(1 + \tfrac{\sigma_x^2}{\sigma_n^2}\lambda_{\max}\Bigr)
> \le
> \tfrac{K}{2}\log\Bigl(1 + c\mathrm{SNR}\Bigr),
> $$
> where $\mathrm{SNR}= \tfrac{\sigma_x^2}{\sigma_n^2}, \qquad c = \lambda_{\max},$ and $c$ is a constant independent of $d_h$. Dividing by $d_h$ yields
> $$
> \frac{I(X;Y)}{d_h}
> \le
> \frac{K}{2d_h}\log\bigl(1 + c\mathrm{SNR}\bigr),
> $$
> showing that average leakage per coordinate scales proportionally to anchor fraction $K/d_h$ (up to $\log(1+\mathrm{SNR})$ factor). In particular, as $K/d_h \to 0$ (for fixed $\mathrm{SNR}$ & bounded $\lambda_{\max}$), the per-coordinate mutual information $I(X;Y)/d_h \to 0$, formalizing our claim that leakage can be made arbitrarily small by keeping $K \ll d_h$.

---

> ### Author Response · Authors · 2025-11-17
> **(Part 3/6) Response to the Reviewer: "The other main weakness of the paper is that it lacks a well defined formal..."**
>
> **(3) How we compare privacy in practice (via Experiments) : attacks and ambiguity vs $K$:**
>
> **Ambiguity / solution-space behaviour vs. $K$:**
> Fig.3 provides empirical support for our geometric claim that reconstruction ambiguity shrinks as the anchor count $K$ approaches the ambient dimension $d_h$. For each dataset, we plot the Frobenius reconstruction error in the recovered high-dimensional embeddings ($X_{\mathrm{frob}}$) under non-IID, unbalanced client partitions. Here, $X_{\mathrm{frob}} = 0$ would correspond to exact recovery of the original high-dimensional configuration, while larger values indicate that many configurations remain compatible with the observed distances. The red vertical line marks the theoretical threshold at $d_h - 1$ (e.g., $783$ for MNIST, $19$ for German Credit), which separates the subcritical regime $K < d_h$ from the regime where exact recovery may become possible.
>
> Empirically, for $K < d_h$ (to the left of the red line), $X_{\mathrm{frob}}$ decreases only gradually and remains bounded away from zero. For Retina and Pneumonia, the theoretical threshold lies outside the plotted $K$-range, so Fig.3 only shows the monotone improvement within the subcritical regime $K < d_h$; in these cases, our theory predicts that a similar collapse in ambiguity would occur at larger $K$.
>
> **Curated Example to Demonstrate Privacy:** We created a curated dataset with private attributes such as age & categorical features like gender or occupation. The setup includes: 5 clients (denoted $C_1$ to $C_5$), Each with 5 attributes (i.e., $d_h=5$ ) & 4 reference anchors (i.e., $K=4<d_h$). The SENSE framework is used to compute embeddings using only client-to-anchor distances. The results are already provided in Table15 in Appendix A.16. We are not providing again here due to space constraints. It is evident (from the results in Table15) that the embeddings generated by the SENSE framework do not expose any sensitive details of the original data.
>
> **Membership-inference and inversion attacks:**
> (a) Model inversion attacks:  These attacks aim to reconstruct input data (e.g., faces, records) by optimizing over overtrained models or partial layers, supervised predictions (logits, class scores), gradients, or overfitted behaviors.  SENSE bypasses all of these attack vectors: no supervised model is trained at the server, and no labels, logits, or gradients are exposed.  The server only observes numeric distance values, i.e., a compressed geometric abstraction of the data.  Thus, standard inversion attacks lack a viable attack surface in our setting.
>
> (b) Membership inference attacks (MIA): MIA strategies typically rely on changes in model confidence when queried with seen vs. unseen samples.  However, SENSE is model-free at the server: it only observes static distance vectors and no prediction scores.
> There is therefore no model behaviour to exploit.  While the standard assumptions behind MIA do not hold here, we still simulate such attacks on SENSE for empirical completeness.
>
> Empirical attack simulations: We test both MIA and inversion attacks across four datasets.
>
> MIA goal: Determine whether a point was part of the client dataset. We extract statistical features (mean, standard deviation, minimum, maximum, quantiles) from distance vectors and train classifiers (MLP, random forest) to predict membership.
>
> Inversion goal: Reconstruct an input point $x$ from its distances $d_i \approx \\|x - a_i\\|$ to anchors. We solve the following optimization problem: $\hat{x} = \arg\min_{x} \sum_{i=1}^K \bigl(\\|x - a_i\\| - d_i\bigr)^2 .$
>
> Evaluation metrics: We assess privacy using 3 metrics: MIA Accuracy, Pixel Error, and Reconstruction Accuracy $(1 - \text{Pixel Error}).$ MIA Accuracy measures the success of membership inference; values near $(50\\%)$ indicate strong privacy.  Pixel Error measures the average deviation between reconstructed & original inputs; higher values mean better protection. We categorize privacy into three levels: HIGH: MIA $(\le 51\\%)$, Pixel Error $(\ge 20\\%)$, MEDIUM: MIA $(48\text{--}51\\%)$, Pixel Error $(15\text{--}20\\%),$ LOW: MIA $(< 48\\%)$, Pixel Error $(< 15\\%)$. These thresholds offer a compact, interpretable view of privacy risk across attacks.
>
> Table1:
>
> |Data|MIA Acc ($\\%$)|Pixel Error ($\\%$)|Privacy Level|
> |-|-|--|-|
> |MNIST|50.5|26.7|High|
> |FMNIST|51.0|24.8|High|
> |BreastMNIST|48.0|14.4|Med|
> |BloodMNIST|48.7|15.8|Med|
>
> Datasets with higher input dimensionality & well-spanned anchor sets $(K)$ exhibit greater resilience to reconstruction, which is consistent with our geometric analysis.  Across all four datasets, MIA accuracy lies in the range $(48\\%)$–$(51\\%)$, i.e., close to random guessing indicating strong protection against membership inference. For inversion the average pixel reconstruction error is $(20.42\\%)$; even in the lowest-error cases, reconstructions are visually degraded, confirming the underconstrained nature of the inversion problem in our setting.

---

> ### Author Response · Authors · 2025-11-17
> **(Part 4/6) Response to the Reviewer: "The other main weakness of the paper is that it lacks a well defined formal..."**
>
> **SENSE enforces privacy by design:** raw features are never shared. Clients only send partial, structured distance vectors which are disjoint and never centralized. This prevents any adversary from reconstructing the global data geometry or latent structure. Even with $K < d_h$, the adversary lacks enough information to infer client embeddings.
>
> **Structure-Aware Adversaries:** A sophisticated adversary could, in principle, exploit intrinsic structure; e.g., if embeddings lie on a low-rank subspace of rank $r$, then the underdetermined system $A x = b$ might become solvable using structured regularization such as nuclear-norm minimization for low-rank or LASSO for sparsity. Prior work shows that privacy can break when both structured distances and strong geometric priors are available; for example, in [3] it is shown that exact recovery with $r+1$ anchors is possible via convex solvers, and nuclear-norm minimization can recover full distance matrices under standard assumptions. In other words, if one simultaneously reveals too many structured distances and the data exhibit strong low-rank or sparse structure, privacy can be at risk.
>
> **Proposed defenses:**
>
> (a) Local Differential Privacy (LDP). We apply LDP to each client’s distance (DNA) vector before sending it. Clients first clip distances to a threshold $\delta$ and then add Laplace noise with scale $\lambda$, so that the released vector is $\tilde{d} = \mathrm{clip}(d, \delta) + \eta$ with $\eta \sim \mathrm{Laplace}(0,\lambda)$, yielding an $\varepsilon$-LDP guarantee with budget $\varepsilon = 2\delta / \lambda$.
>
> (b) Rank-aware anchor capping. If the server receives clients’ estimated ranks $r_1, r_2, \dots, r_N$, it sets the number of anchors to $K = \min(r_1, r_2, \dots, r_N) - 1$. This ensures that no client observes more than $r$ affinely independent anchors, keeping the system underdetermined and preserving privacy even in the presence of low-rank structure.
>
> We test both defenses against structure-aware adversaries on synthetic data. The data are $X_{\mathrm{struct}} = X_{\mathrm{low\text{-}rank}} + 0.5 \cdot X_{\mathrm{sparse}}$, with size $n \times d$, rank $r = 50$, and $d = 100$. We consider three attacks: (i) a low-rank (SVD-based) reconstruction attack, (ii) a sparse (LASSO-based) attack, and (iii) a baseline attack using the anchor feature mean. We report three metrics: RE: reconstruction error, FS: F1-score (true vs. reconstructed), PL: privacy leak $1 - \mathrm{RE} / \mathrm{RE}_{\mathrm{baseline}}$, and mark a run as vulnerable when $\mathrm{RE} \le 0.1$. Table2 (below) shows that, on synthetic low-rank + sparse data, attacks fail to leak private data when LDP is applied, even when the number of anchors satisfies $K \ge r$. As $K \to r$ or $K > r$, adversaries can exploit affine structure: low RE and high FS indicate successful reconstruction (i.e., a privacy leak). However, LDP nullifies leakage even for $K \ge r$. When $K < r$, the system remains underdetermined and we observe no privacy risk. This validates our two defenses. In this setting, low RE and high FS indicate adversary success, whereas high RE or low FS indicate that SENSE remains strongly private.
>
> Table2: On synthetic low-rank + sparse data, attacks fail to leak private data when LDP is applied even when $K \ge r$.
>
> |K|LDP|RE|FS|PL|Vulnerable|
> |-|-|-|-|-|-|
> |49|No|0.9976|0.0318|0.0154|No|
> |50|DP ($\alpha$=0.02, $\delta$=0.0005)|1.0105|0.0445|0|No|
> |80|No|0.4705|0.3675|0.5316|Some|
> |99|No|0.0977|0.9046|0.9029|Yes|
> |99|DP ($\alpha$=0.015, $\delta$=0.0005)|1.0064|0.0450|0|No|
>
> **More attacks on SENSE vs. baselines:** We simulate adversarial conditions (Gaussian noise directly to raw features i.e., pixel-level perturbation of MNIST data): A1: $60\\%$ clients adversarial, all samples. A2: $60\\%$ clients, $30\\%$ samples adversarial. Results are in Table 3 below: For metric definitions: CA = Classification Accuracy with kNN, NMI = Normalized Mutual Information, SC = Silhouette Coefficient, and Overlap@k = Neighborhood Overlap for top-k. We compare SENSE vs FedTSNE [4] and SENSE +attacks vs. FedTSNE+attacks performance:
>
> Table 3:
>
> |Metric|SENSE|FedTSNE|SENSE+A1|FedTSNE+A1|SENSE+A2|FedTSNE+A2|
> |-|-|-|-|-|-|-|
> |CA-1NN|0.93| 0.90|0.88|0.87|0.90|0.88|
> |-10NN|0.92|0.91|0.88|0.87|0.90|0.89|
> |-50NN|0.91|0.89|0.87|0.86|0.88|0.87|
> |NMI|0.67|0.69|0.63|0.60|0.64|0.64|
> |SC|0.44|0.47|0.46|0.44|0.43|0.44|
> |Overlap@1|0.54|0.46|0.28|0.28|0.39|0.32|
> |@10|0.11|0.09|0.06|0.06|0.08|0.07|
> |@50|0.03|0.03|0.02|0.02|0.02|0.02|
>
> References:
> [1] Cover, Thomas M. Elements of information theory.1999.
> [2] Tse, David, and Pramod Viswanath. Fundamentals of wireless communication. 2005
> [3] Lichtenberg, Samuel, and Abiy Tasissa. "Localization from structured distance matrices via low-rank matrix recovery." (2024).
> [4] Qiao, Dong, Xinxian Ma, and Jicong Fan. "Federated t-sne and umap for distributed data visualization." Proceedings of the AAAI Conference on Artificial Intelligence. Vol. 39. No. 19. 2025.

---

> ### Author Response · Authors · 2025-11-17
> **(Part 5/6) Response to the Reviewer: "The paper also does not give a very clear description of the proposed algorithm. E.g. there is no algorithm block....""**
>
> **A3:** We appreciate the reviewer’s concern about the clarity and formalization of the algorithm and guarantees. Below we address each part of the question in turn.
>
> **1. Algorithm description and unifying objective:** Where is the algorithm described? Appendix A.6: step-by-step pseudocodes for SENSE, from local data to the final low-dimensional embedding. In the revision, we will move a compact algorithm block into the main text (from App. A.6) and explicitly group the steps as discussed in **A2**.
>
> **2. Precise assumptions and computational problems:** We explicitly assume: There are $M$ clients and one server. Each client $m$ holds a disjoint index set $\mathcal I_m$, and $\\{\mathcal I_m\\}_{m=1}^M$ forms a partition of $\\{1,\dots,N\\}$. Each non-anchor (NA) point lives in $\mathcal M \in \\{\mathbb R^{d_h}, \mathbb H^{d_h}\\}$. The server has a small public anchor set $P = \\{p_1,\dots,p_K\\}$ with $\mathrm{rank}(P) = K < d_h$ (for privacy). The observation mask $\Omega \subseteq \\{1,\dots,K+N\\}^2$ encodes which entries of $D$ are observed under Pointwise/Multisite $\times$ Full/Partial (Sec.3, Fig.1).
>
> Computational problems: Stage 1 (matrix completion in the distance domain): Given $\Omega$ and the observed entries $\\{D_{ij}\\}$ where ${(i,j) \in \Omega}$, find a high-dimensional latent embedding $X$ such that $D(X)$ fits the observed entries in Frobenius norm: $\min_X \;\sigma(X)=\big\\|P_\Omega\big(D(X) - D\big)\big\\|_F^2.$ Stage 2 (DR on reconstructed geometry): Given a complete distance matrix, minimize a standard NE/contrastive DR loss to obtain low-dim embeddings.
>
> **3. Reconstruction error: natural scale, bounds, and dependence on $K$ and $N$:** Below we summarize the formal guarantees that support the reconstruction step; these results are standard in the distance-matrix completion literature and we will include them explicitly in the revision.
>
> Clarification: Our formal guarantees apply only to privacy: Thms.3.1 \& 3.2 show that when $K < d_h$, exact recovery of features from anchor-NA distances is impossible. Regarding fidelity we convey that as $K \to d_h - 1$, SMACOF preserves relative neighborhood structure not raw features. This reconstruction is consistent up to Euclidean isometries (translation, rotation) and potential global scaling which are standard ambiguities in metric embedding methods such as classical MDS and SMACOF. In privacy-preserving settings this is sufficient for downstream tasks. We support this empirically (Fig.3).
>
> (a) Natural scale of reconstruction error [6]: The magnitude of the error in Eq.(5) is governed by classical results on Euclidean distance matrix (EDM) completion. Let $D^\star$ be the true EDM, $\hat D$ the reconstruction, and assume rank $r$ and incoherence parameter $\mu$. Under standard matrix-completion conditions, one has a bound of the form $\mathbb{E}\big\\| \hat D - D^\star \big\\|_{F^2}  \le C \frac{r \mu N \log^2 N}{|\Omega|},$ which quantifies the natural scaling of the error with the intrinsic rank, the number of observed distances $|\Omega|$, and hence indirectly with the number of anchors $K$ (since $|\Omega| \propto K N$ in our setting).
>
> (b) Error bounds under SVD-based localization [2]: When reconstruction is performed via SVD-based solvers (a stronger alternative to SMACOF), one obtains bounds of the form $\big\\| D - S_4 \big\\| = O\\!\left( n d_{\max}^4 + n^{3/2} d_{\max}^3 \right),$
> where $d_{\max}$ denotes the maximum inter-point distance. This gives explicit dependence on geometry through $d_{\max}$.
>
> (c) Expected EDM error [1-5]: For EDM completion with $m$ observed distances, rank $r$, and noise parameters $(\zeta,\nu)$, a typical guarantee is $\frac{1}{n} \mathbb{E}\big[\\|D - D_F\\|_F\big] \le C\sqrt{\frac{r}{m}}(\zeta + \nu),$ showing that fidelity improves as the number of observations $m$ grows (which, in SENSE, is controlled by the anchor count $K$ and the number of clients $N$).
>
> Note: Notably, the quality of distance-based reconstruction using different solvers is known to depend on the geometry & condition of observed distances. Our choice of SMACOF was guided by ease of implementation & its compatibility with distance-based privacy, but we acknowledge that stronger theoretical guarantees are available via other solvers as provided above.
>
> References:
> [1] P. Drineas et.al., Distance matrix reconstruction from incomplete distance information for sensor network localization. 2006.
> [2] H. Zhang, et.al., Localization from incomplete euclidean distance matrix: Performance analysis for the svd–mds approach. 2019.
> [3] S. Lichtenberg and A. Tasissa. A dual basis approach to multidimensional scaling, 2024a.
> [4] K.i V. Mardia and A. D. Riley. The classical multidimensional scaling revisited, 2021.
> [5] Usman A Khan et.al., Distributed sensor localization in random environments using minimal number of anchor nodes. 2009.
> [6] Candes, Emmanuel et.al., "Exact matrix completion via convex optimization." (2012).

---

> ### Author Response · Authors · 2025-11-17
> **(Part 6/6) Response to the Reviewer: "The paper's claim about efficiencuy are also very unclear..... and also In terms of novelty..."**
>
> **A4:**  As discussed in Appendix.A.16: With $N$ non–anchor (NA) points and $K$ anchors, the anchored-MDS solver in Sec.2 operates only on anchor-NA and (optional) local NA-NA distances. Its per-iteration cost is $\mathcal{O}\big(K^2 d_h + K N d_h\big),$ since all updates involve $K$-dimensional anchor rows and $N$ NA points; the full $N\times N$ raw distance matrix is not estimated in this stage as private features are not shared. In all experiments we work in the privacy regime $K \ll d_h$ and $K \ll N$, so the dominant term is $\mathcal{O}(K N d_h)$, i.e., linear in $N$, comparable to distributed PCA/JL which scales as $\mathcal{O}(N d_h k)$ for $k$ components. After this step, the server has reconstructed embeddings $\hat X \in \mathbb{R}^{N \times d_h}$ and may compute NA--NA distances from $\hat X$ locally; this does not require any further communication.
>
> Low-dimensional embedding stage: We use the fastest variants of different global low-dimensional embedding methods in our framework, as this is the second stage in the pipeline. Notation: here k denotes the number of neighbors considered per point (in the attractive force calculation), $d_l$ is the embedding dimension, $n$ are the number of data points (samples) and $m $ are the number of negative (repulsive) samples per positive interaction. 1) Fast NE methods: Van-t-SNE takes $\mathcal{O}(n^2 d_l)$ [1], Barnes-Hut t-SNE (BH-t-SNE) and FIt-SNE (FFT-based interpolation) takes $\mathcal{O}(kn \log n \cdot 2^{d_l})$ and $\mathcal{O}(kn \cdot 2^{d_l})$ respectively. 2) The contrastive neighbor embedding: These methods only sample $nm$ repulsive interactions per epoch (instead of all pairs) with complexity $\mathcal{O}(kmnd_l)$, which  scales linearly with embedding dimension $d_l$. This also includes NC-t-SNE/UMAP with contrastive loss runs in $\mathcal{O}(kmd_l)$ with $m \ll n$ repulsive samples per epoch [2].
> Putting both stages together, the end-to-end complexity of SENSE is $\text{Reconstruction: } \mathcal{O}(K N d_h), \quad \text{Contrastive DR: } \mathcal{O}(k m N d_l),$ so the pipeline scales approximately linearly in the number of points $N$.
>
> Communication rounds: SENSE also differs from federated PCA/JL in its communication pattern. In SENSE, clients send their anchor-distance vectors (and optional local NA-NA distances) to the server in a single non-interactive round; all subsequent reconstruction and DR steps run entirely at the server. In contrast, federated PCA and related projection-based methods typically require multiple communication rounds to iteratively estimate the global covariance or subspace. So intuitively we can say, under a stricter privacy model (no raw features or linear projections ever leave the client) SENSE achieves approximately PCA/JL-like $\mathcal{O}(N)$ time with only one round of communication. We will make this two-stage complexity and one-shot communication protocol explicit in the revised version.
>
> **A5:** We respectfully disagree that the contribution is unclear. To our knowledge, SENSE is the first framework that: (i) formulates decentralized neighbour-embedding as a structured distance matrix completion problem with anchors under four visibility regimes; (ii) provides a geometry-aware anchored-MDS / hyperbolic solver that can be plugged into any Euclidean or hyperbolic DR objective; and (iii) analyses a geometric non-identifiability guarantee for decentralized DR in terms of the number of anchors versus embedding dimension, supported by extensive ablations and reconstruction-attack experiments. This positions SENSE at the intersection of DR, decentralized learning, and geometric ML, and we believe it will be of interest to other communities as well.
>
> Also, as mentioned in the Introduction section of the paper: these properties make SENSE broadly applicable to privacy-sensitive, structurally diverse domains. Hospitals can jointly visualize patient data without violating HIPAA/GDPR, banks can detect fraud patterns without exposing transactions, and even mobile/IoT devices with a single sample can contribute to global embeddings. Genomic labs can embed single-cell transcriptomes into a shared hyperbolic space that preserves both cellular hierarchy and privacy. Crucially, SENSE also supports dynamic participation: new clients or samples can be incorporated by estimating partial distances to existing entities, avoiding full re-computation while maintaining global coherence. Thus, SENSE is not only privacy-preserving and geometry-aware but also inherently scalable to dynamic federated ecosystems.

---

> ### Author Response · Authors · 2025-11-22
> **Requesting Reviewer o7FR**
>
> Dear Reviewer o7FR,
>
> Thank you again for your detailed and thoughtful feedback on our submission. We have provided a point-by-point rebuttal in order to address the raised concerns.
>
> As the rebuttal period is ending in a few days, we would be very grateful if you could kindly let us know whether any key concerns remain or if our clarifications address your main objections. If the rebuttal alleviates some of your worries, we would also appreciate it if you could consider updating your score accordingly.
>
> Thank you for your time and consideration.
>
> Sincerely,
>
> Authors.

---

> > ### Author Response · Authors · 2025-11-27
> > **A gentle reminder**
> >
> > Dear Reviewer o7FR,
> >
> > Thank you again for your detailed and thoughtful feedback on our submission. We have provided a point-by-point rebuttal in order to address the raised concerns.
> >
> > As the rebuttal period is ending in a few days, we would be very grateful if you could kindly let us know whether any key concerns remain or if our clarifications address your main objections. If the rebuttal alleviates some of your worries, we would also appreciate it if you could consider updating your score accordingly.
> >
> > Thank you for your time and consideration.
> >
> > Sincerely,
> >
> > Authors.

---

> > > ### Comment · Area_Chair_ZLdp · 2025-11-29
> > > **To Reviewer o7FR**
> > >
> > > Has the authors addressed your concerns? Please respond with an updated score and/or rebuttal to allow the authors to know your justification.

---

### Official Review · Reviewer_qbSV · 2025-11-01

**Soundness:** 3
**Presentation:** 3
**Contribution:** 2
**Rating:** 4
**Confidence:** 3

**Summary:**

The paper proposes SENSE, a geometry-aware framework for decentralized dimensionality reduction that preserves privacy while maintaining structural relationships in data. Instead of sharing raw features, SENSE uses distances to anchor points as coordination signals, enabling faithful low-dimensional embeddings under privacy constraints. The framework supports both Euclidean and hyperbolic spaces and operates across four data observation regimes. The authors claim provable privacy guarantees without relying on traditional mechanisms like differential privacy or homomorphic encryption, demonstrating competitive performance with centralized baselines while preserving privacy.

__Code link is accessible but files are not__

**Strengths:**

__Novel privacy approach__: The paper makes a significant contribution by framing privacy through geometric underdetermination rather than traditional noise injection. As stated in the paper, "_geometric underdetermination preserves both fidelity and privacy_" (p.4), offering a conceptually fresh perspective that avoids the utility-privacy trade-off inherent in differential privacy approaches. This represents genuine originality in privacy-preserving representation learning.

__Practical deployment flexibility__: SENSE demonstrates impressive adaptability across multiple scenarios, handling both Pointwise (single non-anchor per client) and Multisite (multiple non-anchors) settings. The framework's ability to work with "four observation regimes" reflecting "_real-world data availability_" (p.3) enhances its practical significance for real-world applications like healthcare and finance where data distribution constraints are common.

**Weaknesses:**

__Unjustified complexity over simpler alternatives__: The paper doesn't sufficiently justify why its matrix completion approach is preferable to adding differential privacy to standard dimensionality reduction. As the authors note "injected noise quickly degrades embedding quality" (p.4), but they don't quantitatively compare against modern DP mechanisms that carefully calibrate noise to preserve utility. For large-scale applications, the matrix factorization complexity ($O(n^3)$ in worst case) creates unnecessary computational burden compared to simpler DP approaches.

__Insufficient scalability analysis__: While Table 13 mentions "scalability and runtime on large-scale datasets," the paper lacks concrete evidence of how SENSE performs on truly massive datasets (millions of points). The Tiny ImageNet and SVHN experiments don't demonstrate whether the approach remains practical at web-scale.

__Incomplete privacy analysis__: The claim of "provable reliability" with "formal privacy guarantees" lacks rigorous mathematical treatment. The discussion about anchor count (K) affecting privacy (p.21) needs formal privacy bounds rather than empirical observations. Without proper privacy accounting, it's difficult to compare against established privacy frameworks like differential privacy.

**Questions:**

1. How would SENSE's utility-privacy trade-off quantitatively compare against adding calibrated Gaussian noise (DP-SGD) to standard t-SNE/UMAP? Could you provide $\epsilon, \delta$ values for your approach?
2. For $K < d$ anchors, what's the exact mathematical relationship between K, dimensionality d, and the privacy leakage? Can you derive a formal privacy bound similar to differential privacy's $\epsilon$?

---

> ### Author Response · Authors · 2025-11-22
> **(Part 1/3) Response to the Reviewer: W1. "Unjustified complexity over simpler alternatives."**
>
> > **Code link is accessible but files are not:**
>
> **We have re-checked the code repository link and confirmed that the files are publicly accessible (we also verified this from multiple independent accounts). To eliminate any remaining access issues, we now additionally include the same code and files in the supplementary material.**
>
> Ans. We thank the reviewer for raising this comparison. Our goal is not to claim that SENSE is universally simpler than all DP methods, but to clarify that it solves a problem under a different system model. In SENSE, each client computes distances from its NAs to a small anchor set & (optionally) a few local NA–NA distances & sends these once to the server. All subsequent steps (matrix completion & low-dim embedding) are performed centrally on the server. There is no iterative gradient exchange or multi-round communication.
>
> However, modern DP approaches for t-SNE/UMAP require repeated, noisy gradient updates, each with clipping, Gaussian noise & privacy accounting. In FL this translates into many communication rounds as in Fed-tSNE [1] / Fed-UMAP & their DP variants. Thus, while the per-iteration cost may look simple the overall communication cost is substantial. We compare & provide empirical results (along with runtime) of SENSE vs fedtsne & report different metrics such as CA (kNN Classification Accuracy), NMI (Normalized Mutual Info), SC (Silhouette Coef.), & Overlap@k.
>
> **Table1:**
> |Method|NA|Anchors|time(s)|1-NN|10-|50-|NMI|SC|Overlap@1|@10|@50|
> |-|-|-|-|-|-|-|-|-|-|-|-|
> |SENSE|1k|100|70.27|0.69|0.66|0.67|0.62|0.46|0.28|0.08|0.03|
> |||783|76.16|0.72|0.73|0.71|0.58|0.47|0.45|0.11|0.04|
> ||2k|100|79.56|0.71|0.75|0.69|0.59|0.47|0.22|0.06|0.02|
> |||783|97.26|0.76|0.74|0.71|0.62|0.47|0.43|0.10|0.03|
> |fedtsne|1k|100|147.78|0.66|0.71|0.66|0.55|0.48|0.35|0.09|0.03|
> |||783|139.40|0.67|0.65|0.60|0.58|0.48|0.47|0.12|0.04|
> ||2k|100|146.71|0.72|0.70|0.65|0.56|0.49|0.24|0.07|0.02|
> |||783|151.36|0.74|0.72|0.68|0.57|0.48|0.39|0.09|0.03|
>
> SENSE outperforms FedTSNE on both runtime & acc. Runtime includes full pipeline: anchor generation+matrix completion+t-SNE. Also see response to **Reviewer o7FR (Part 4/6) Table 3.**
>
> Additionally, we perform experiments & report the NMI scores: Instead of sending raw data to server the traditional FL setting applies Local Differential Privacy (LDP) to the parameters. It involves 2 hyperparameters, $\delta$ which is the clipping threshold & $\lambda$ which is the Laplacian noise parameter. We apply LDP on the raw features & present the NMI metric to evaluate the performance of SENSE with different $\delta$ & $\lambda$ values.
>
> **Table2:**
> |Data/ λ|δ|Without LDP|0.002|0.005|0.01|
> |-|-|-|-|-|-|
> |Iris|δ = 0.2|0.892|0.8106|0.6946|0.6183|
> ||δ = 0.1||0.7169|0.6138|0.5060|
> |Seeds|δ=0.2|0.701|0.4367|0.3605|0.1828|
> ||δ = 0.1||0.3383|0.0430|0.0112|
>
> **Table3:**
> ||λ=0.0025|λ=0.005|λ=0.01|
> |-|-|-|-|
> |ε for (δ = 0.2)|160|80|40|
> |ε for (δ = 0.1)|80|40|20|
>
> We observed: use of a higher value of $\lambda$ can result in decrease in performance. Because increasing value of $\lambda$ results in stronger Laplace noise being added to raw features, which ultimately leads to less accurate performance. Also, by considering the upper bound on the privacy budget, which is: $\epsilon = \frac{2 \delta}{\lambda}$, Table3 shows that as $\lambda$ increases & $\delta$ decreases the privacy budget $\epsilon$ becomes smaller, indicating better privacy protection but with a significant impact on utility.
>
> We also extend SENSE & apply DP on top of it to ensure more privacy against different structure-aware adversaries & explain how these ((Part 4/6) Reviewer qbSV response) might affect SENSE & how we can use DP to cater it. But here also the DP based solution leads to utility degradation when used along with SENSE. For full details see **Reviewer qbSV Table2.**
>
> Morever, we empirically test **Why SENSE Avoids Noise-Based Privacy in general:** While we acknowledge that such approaches can be used, the goal of SENSE is fundamentally different. Our objective is to reconstruct or approximate inter-point distances not the raw features while ensuring that high-dim features of NA points remain private. Importantly in SENSE distances are not treated as private. Instead, they are intermediate coordination variables similar to how they are used in classical MDS or sensor network localization. The core idea is to protect the original feature representations, not to hide the distance map itself. That said, we recognize that noise-based privacy mechanisms may be appropriate in some scenarios, particularly where (a) robust denoising resistance is guaranteed & (b) added noise does not critically degrade performance. However, we chose not to use noise-based methods in SENSE for the following reasons: Empirical Evaluation and detailed discussion are in Table 16 and 17 in Appendix A.16, page no. 29.
>
> References:
> [1] Qiao et.al. Federated t-sne and umap for distributed data visualization. 2025.

---

> ### Author Response · Authors · 2025-11-22
> **(Part 2/3) Response to the Reviewer: W2 and W3:**
>
> > **W2: Insufficient scalability analysis:**
>
> **Ans:** We thank the reviewer for pointing out the need to clarify scalability. Below we clarify this:
>
> (1) To our knowledge, the DR literature (especially for non-linear methods like t-SNE/UMAP) typically operates on much smaller datasets than large-scale classification FL, as also noted in [1]. This is largely because these methods optimize non-convex objectives & their computational cost remains substantial even in the centralized setting. In line with this practice, we benchmark SENSE on the standard datasets used in DR & federated DR & already go beyond the scales explored in prior decentralized DR methods [1-3]. In particular, SENSE is evaluated up to $\sim 9\times 10^4$ points. While we have not yet included million-point web-scale experiments, we expect SENSE to be applicable in realistic deployments. Crucially, the different SENSE setups naturally cover both cross-device & cross-silo regimes.
>
> (2) Asymptotic complexity: See (Part 6/6) Response to the Reviewer: A4 also check Appendix (Table 14).
>
> (3) Also see Table1 (Part 1/3): SENSEvs federated DR baseline FedTSNE. SENSE is roughly faster in run-time as compared to FedTSNE for these fed DR tasks, even though FedTSNE does not include our matrix-completion step.
>
> (4) Matrix-completion bottleneck and stochastic MDS as a path to larger scales: Our current matrix-completion stage uses an MDS-SMACOF solver. Classical SMACOF becomes expensive as $N$ grows, because each iteration applies global updates involving essentially all $O(N^2)$ pairwise distances. The stochastic MDS framework of [4] proposes a stochastic SMACOF algorithm that processes only a sparse subset of distances at each iteration. Concretely, the $N$ points are partitioned into clusters of size $p$; if each cluster Laplacian has $q \ll p^2$ non-zero weights, then one iteration processes $f(N) =\frac{N}{p}q$ distances and has per-iteration complexity: $O\bigl(f(N)\log p\bigr),$ i.e., near-linear in the number of distances touched per step, up to a logarithmic factor in the cluster size $p$. Equivalently, if we denote by $E_t$ the set of distance pairs used at iteration $t$, the per-iteration cost is on the order of: $O\bigl(|E_t|d_\ell\bigr)$ (up to a $\log p$ factor), where $d_\ell$ is the low-dimensional embedding dimension. In [4], it is proven that the stochastic trajectory tracks an averaged algorithm that converges to a stationary point of the (mean) stress function and empirically demonstrates scalability on large sensor-network localization and visualization tasks.
>
> Plugging such a stochastic MDS solver into SENSE would (i) avoid repeatedly forming and updating the full $K\times N$ distance block, (ii) enable streaming or mini-batch processing of anchor-NA distances, and (iii) reduce the practical constants in the Stage-2 complexity while preserving the same stress objective and privacy mechanism. Technically, SENSE is fully compatible with these stochastic SMACOF-type solvers and can therefore be pushed to larger $N$ without changing the underlying geometric privacy model.
>
> (5) Also, we experiment on Random Geometric Graphs (RGG). We report FS, RE, & runtime $t$ (sec):
>
> |N|d|K|FS|RE|time (s)|
> |-|-|-|-|-|-|
> |500|100|99|0.79|0.020|16|
> |||90|0.78|0.030|13|
> |||50|0.64|0.040|5|
> |||30|0.51|0.052|2|
> ||700|699|0.82|0.014|660|
> |||350|0.78|0.019|312|
> |||100|0.72|0.023|30|
> |||50|0.65|0.038|10|
>
> SENSE offers a continuous knob between accuracy & computational cost via the anchor budget $K$: inc $K$ improves reconstruction & DR quality but inc the $O(K N d_h)$ cost, while dec $K$ reduces runtime & communication at the expense of some fidelity. In our main experiments we choose $K < d_h$ both for privacy & to keep runtime controlled, but the RGG results show that the method remains usable even when $K \ll d_h$, which is precisely the regime one would adopt at very large scales.
>
> > **W3: Incomplete privacy analysis:**
>
> **Ans:** Please check *(Part 2/6) Response to the Reviewer: "The other main weakness of the paper is that it lacks a well defined formal..."*
>
> References:
> [1] Qiao et.al. Federated t-sne and umap 2025.
> [2] Li, Ziwei, et al. FEDNE 2024.
> [3] Saha et al. "Federated, fast, and private visualization of decentralized data." 2023.
> [4] Rajawat et.al. "Stochastic MDS." 2017.

---

> ### Author Response · Authors · 2025-11-22
> **(Part 3/3) Response to the Reviewer: Questions1 and 2**
>
> > Q1: How would SENSE's utility-privacy trade-off...?
>
> **Ans1:** Our privacy mechanism in SENSE is geometric, not DP-SGD-based. Privacy arises from under-determined anchor–distance sharing: when $K<d_h$, the anchor–distance map is non-invertible and high-dimensional features are non-identifiable. Because the basic SENSE protocol is fully deterministic, clients send exact squared distances to the server: there is no inherent randomness, and hence no natural $(\epsilon, \delta)$ DP characterization unless an explicit DP mechanism is added on top. In principle, one could run DP-SGD on top of SENSE or on a federated t-SNE/UMAP objective, but doing this rigorously is non-trivial.
>
> Existing work shows that practical DP-SGD guarantees are highly sensitive to implementation details such as Poisson vs. shuffled sampling and the choice of privacy accountant and that seemingly innocuous choices can substantially mis-estimate $(\epsilon, \delta)$ [1], so we avoid presenting a “quick-and-dirty” DP-SGD baseline without a careful, dedicated analysis. To our knowledge, there is currently no DP-SGD–based federated t-SNE/UMAP implementation; available systems such as FedTSNE already operate near the edge of computational feasibility even without full DP accounting. FedTSNE does provide DP variants, which we discuss in detail in (Part 1/3), with corresponding results in Table 1.
>
> Given this landscape, in this paper we focus on comparisons that are actually available and well-defined in the literature: For a complete SENSE–DP discussion and results, we refer the reviewer to the full answer in (Part 1/3).
>
> We therefore view DP / DP-SGD as complementary to our geometric privacy mechanism rather than as a competing baseline. Adding DP noise on top of SENSE is a natural direction for future work, specifically, to investigate whether one can achieve a good privacy–utility trade-off while retaining the geometric advantages of SENSE.
>
> We fully acknowledge that every privacy mechanism has its own strengths and limitations. While DP, DP-SGD, and noise-based methods offer strong formal guarantees, SENSE deliberately takes a different route: it is grounded in deterministic geometry and tailored to decentralized, structured, and privacy-sensitive scenarios where sharing raw features or gradients is undesirable. We see this as a complementary direction in privacy-preserving representation learning. We sincerely thank the reviewer for these suggestions and view them as promising avenues for future work.
>
> > Q2: For $K < d$ anchors, what's the exact mathematical relationship....?
>
> **Ans2:** This is addressed in detail in *(Part 2/6) Response to the Reviewer: "The other main weakness of the paper is that it lacks a well defined formal...".* Very briefly, we show here:
>
> 1. Geometric non-identifiability: When $K < d_h$, the anchor distance map $\phi(x)$ is non-injective and the preimage of a given distance vector has dimension $d_h-K$, so a continuum of high-dimensional points is consistent with the same distances.
> 2. Information-theoretic leakage: In a local linear Gaussian model, we prove that the per-coordinate mutual information satisfies $$\frac{I(X;Y)}{d_h} \le \frac{K}{2d_h}\log\bigl(1 + c\mathrm{SNR}\bigr),$$ so the average leakage per coordinate scales proportionally to $K/d_h$.
>
> We do not claim a DP-style $(\epsilon,\delta)$ guarantee; instead, Part 2/6 provides these explicit geometric and mutual-information bounds as our formal privacy characterization in terms of $K$ and $d_h$.
>
>
> References:
> [1] Chua, Lynn, et al. "How private are dp-sgd implementations?." arXiv preprint arXiv:2403.17673 (2024).

---

> > ### Author Response · Authors · 2025-11-25
> > **Requesting Reviewer qbSV**
> >
> > Dear Reviewer qbSV,
> >
> > Thank you again for your detailed and thoughtful feedback on our submission. We have provided a point-by-point rebuttal in order to address the raised concerns.
> >
> > As the rebuttal period is ending in a few days, we would be very grateful if you could kindly let us know whether any key concerns remain or if our clarifications address your main objections. If the rebuttal alleviates some of your worries, we would also appreciate it if you could consider updating your score accordingly.
> >
> > Thank you for your time and consideration.
> >
> > Sincerely,
> >
> > Authors.

---

> > > ### Author Response · Authors · 2025-11-27
> > > **A gentle reminder and request**
> > >
> > > Dear Reviewer qbSV,
> > >
> > > Thank you again for your detailed and thoughtful feedback on our submission. We have provided a point-by-point rebuttal in order to address the raised concerns.
> > >
> > > As the rebuttal period is ending in a few days, we would be very grateful if you could kindly let us know whether any key concerns remain or if our clarifications address your main objections. If the rebuttal alleviates some of your worries, we would also appreciate it if you could consider updating your score accordingly.
> > >
> > > Thank you for your time and consideration.
> > >
> > > Sincerely,
> > >
> > > Authors.

---

### Meta-Review · Area_Chair_ZLdp · 2026-01-03

**Summary:**

The recommendation to reject is based on a combination of critical procedural failures in the review process and remaining technical skepticism regarding the paper's core claims. While the authors provided a robust rebuttal, the primary concern lies in the unresolved tension between the proposed "geometric privacy" and established privacy standards (like Differential Privacy). Reviewer o7FR remained fundamentally unconvinced of the motivation and technical exposition as it veers from the standard DP bounds, and Reviewer qbSV raised similar concerns regarding a standard privacy accounting framework. The situation was further complicated by an invalid, mismatched review from Reviewer seDM, leaving the submission without a sufficient number of high-quality, engaged evaluations to justify acceptance.

**Reviewer Concerns:**

* **Privacy Framework Comparability (Outstanding):** The authors introduced a deterministic geometric non-identifiability theorem and an information-theoretic leakage bound. However, the lack of a standardized -Differential Privacy characterization makes it difficult to quantitatively compare SENSE against modern private dimensionality reduction (DR) baselines. The reviewer's concern that geometric underdetermination might not provide sufficient protection against sophisticated adversaries remains a critical point of skepticism.
* **Scalability to Web-Scale Data (Outstanding):** The authors addressed complexity theoretically by suggesting a stochastic MDS solver to achieve  complexity. However, actual experiments on truly massive datasets (millions of points) were not provided. The current empirical evidence remains limited to datasets in the  range, leaving the framework's "web-scale" practicality unverified.
* **Communication Efficiency (Resolved):** The authors effectively demonstrated that SENSE’s "one-shot" protocol is superior to iterative federated learning methods in terms of communication rounds.
* **Mathematical Formalization (Resolved):** The authors successfully formalized the "underdetermination" property.

**Reviewer Scores:**

Likely unchanged.

---

### Decision · Program_Chairs · 2026-01-26

Reject